# Large Continual Instruction Assistant

Jingyang Qiao [1 2]  Zhizhong Zhang [1]  Xin Tan [1 3]  Yanyun Qu [4]  Shouhong Ding [5]  Yuan Xie [1 2]

## Abstract

Continual Instruction Tuning (CIT) is adopted to continually instruct Large Models to follow human intent data by data. It is observed that existing gradient update would heavily destroy the performance on previous datasets during CIT process. Instead, Exponential Moving Average (EMA), owns the ability to trace previous parameters, which can aid in decreasing forgetting. Nonetheless, its stable balance weight fails to deal with the ever-changing datasets, leading to the out-of-balance between plasticity and stability. In this paper, we propose a general continual instruction tuning framework to address the challenge. Starting from the trade-off prerequisite and EMA update, we propose the plasticity and stability ideal condition. Based on Taylor expansion in the loss function, we find the optimal balance weight can be automatically determined by the gradients and learned parameters. Therefore, we propose a stable-plasticity balanced coefficient to avoid knowledge interference. Based on the semantic similarity of the instructions, we can determine whether to retrain or expand the training parameters and allocate the most suitable parameters for the testing instances. Extensive experiments across multiple continual instruction tuning benchmarks demonstrate that our approach not only enhances anti-forgetting capabilities but also significantly improves overall continual tuning performance. Our code is available at https://github.com/JingyangQiao/CoIN.

## 1. Introduction

Large Foundation Modals (LFMs) have demonstrated remarkable capabilities in multi-task understanding and gener-

ation (Li et al., 2023; Zhu et al., 2023; Achiam et al., 2023; Touvron et al., 2023a). It generally exists in two stages: large scale pre-training and instruction-tuning. Instruction tuning is extremely significant due to guiding LFMs (Chen et al., 2024b; Zheng et al., 2023; Touvron et al., 2023b) in following human intent and aligning different modalities (Li et al., 2024; Liu et al., 2023), which seems to be an essential technique to enhance the capabilities and controllability of LFMs *e.g.* LLMs, MLLMs (Jiang et al., 2024).

As knowledge continuously evolves with the development of human society, new instructions are constantly generated, *e.g.* the emergence of new concepts or disciplines. How to enable existing LFMs to assimilate novel instructions and undergo self-evolution becomes the key challenge (Zhu et al., 2024). To accommodate the new instructions, the most effective strategy is incorporating both old and new instructions for joint training. However, even such relatively lightweight fine-tuning is unaffordable. Furthermore, directly fine-tuning these new instructions would destroy the pre-training knowledge, *e.g.* catastrophic forgetting (Goodfellow et al., 2013; Li et al., 2019; Nguyen et al., 2019).

Continual Instruction Tuning (CIT) is proposed to address this challenge (Zhang et al., 2023; He et al., 2023; Chen et al., 2024a). Taking MLLMs as a case study, previous methods, EProj (He et al., 2023), FwT(Zheng et al., 2024), and CoIN (Chen et al., 2024a), utilize the model expansion framework by continually adding new branches for the novel instructions, which therefore has less impact on the old knowledge. However, they suffer from memory explosion and high computational cost problems. On the other hand, continually full fine-tuning (FFT) downstream datasets with single branch architecture would destroy the pre-trained parameters, and greatly reduce the zero-shot generalization performance of MLLMs (Zhai et al., 2023).

Considering the essential mechanism of parameter update, we discover that the gradient update might not be a satisfactory choice for CIT. First of all, we find that the gradient inevitably drives the update of parameters toward the optimization of the new dataset, which causes forgetting. Instead, Exponential Moving Average (EMA) adopts a weighted summation between old and new model parameters, which enjoys the natural advantage of keeping old knowledge. However, it faces the challenge of balancing old and new

---

[1]East China Normal University [2]Shanghai Innovation Institute [3]Shanghai AI Laboratory [4]Xiamen University [5]Tencent YouTu Lab. Correspondence to: Zhizhong Zhang <zzzhang@cs.ecnu.edu.cn>, Yanyun Qu <yyqu@xmu.edu.cn>.

*Proceedings of the 42nd International Conference on Machine Learning*, Vancouver, Canada. PMLR 267, 2025. Copyright 2025 by the author(s).

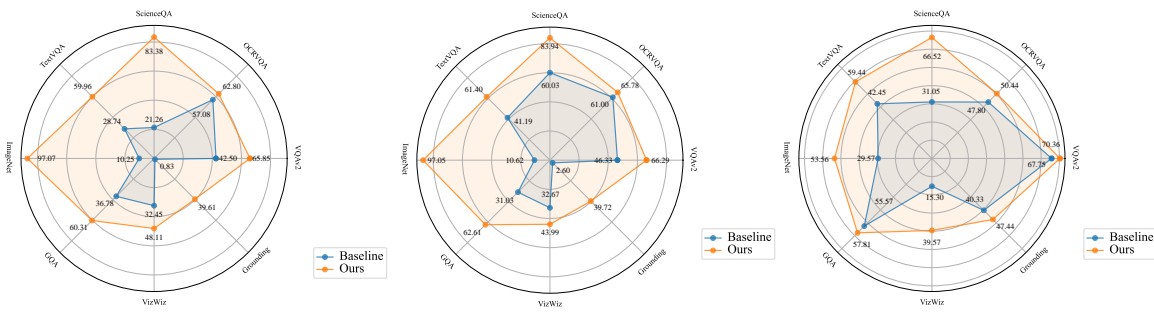

(a) Baseline *v.s.* Ours on LLaVA-7B  (b) Baseline *v.s.* Ours on LLaVA-13B  (c) Baseline *v.s.* Ours on Qwen-VL

*Figure 1.* Radar chart of comparisons on Final Accuracy (higher is better) between baseline (LoRA Fine-Tune) and ours.

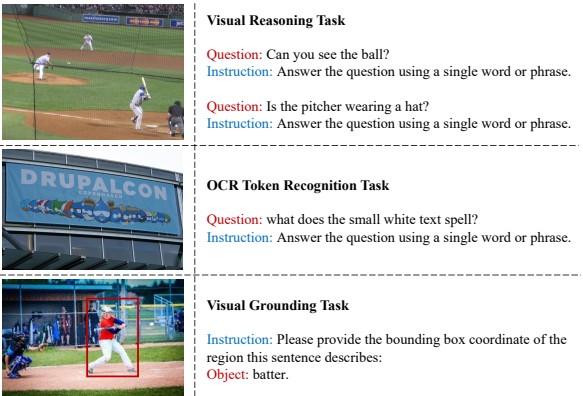

*Figure 2.* Instruction reuse in multimodal instruction datasets.

knowledge in continual tuning, as a fixed EMA weight cannot adapt to the continuously evolving datasets, *e.g.* from location identification task to OCR token recognition task or from sentiment analysis task to question answering task. To determine this balance weight, it seems that the gradient actually represents the magnitude or discrepancy between the model and the new instructions.

In this paper, we propose a general continual instruction tuning framework to address the catastrophic forgetting in CIT. We use Taylor expansion of loss function to formulate the stability and plasticity ideal equations with the EMA update. By employing the Lagrange multiplier method to solve the equations, we provide a theoretical derivation for dynamically updating the weight of EMA in each training iteration, aiming to achieve the optimal balance of stability and plasticity. Furthermore, for different tasks, the instructions may be similar or even identical, as shown in Figure 2. We refer to this phenomenon as instruction reuse. Therefore, we construct a codebook composed of historical instructions and group them according to their semantic similarity. Each group corresponds to a set of trainable parameters. During the training phase, we select the trainable parameters corresponding to the most semantically similar group with the current instruction and perform CIT. It is worth noting that

the number of groups across various instruction templates, is approximately 50% of the number of tasks. Therefore, it can be regarded as a limited model-expansion method.

In order to validate our method, for MLLMs, we adopt LLaVA-1.5 (Liu et al., 2024a) and Qwen-VL (Bai et al., 2023) as the backbone and insert the efficient-tuning parameters: LoRA (Hu et al., 2022) in the LLM. We continually fine-tune on multimodal instruction datasets and verify performances of the tuned model. Furthermore, we also implement our method with the LM-adapted version of T5-small on the NLP continual instruction tuning tasks (Raffel et al., 2020; Zhang et al., 2023). Experimental results consistently demonstrate excellent anti-forgetting, and continual tuning performance of our method. In summary, the contributions of this paper are as follows:

**Alleviating Catastrophic Forgetting in CIT.** Through rigorous deduction, we propose a dynamical EMA weight-adjusting method of reducing catastrophic forgetting in continual instruction tuning.

**Generalized Application and Limited Tuning Costs.** Our method is model-agnostic and can be easily applied to a wide range of CIT methods. Additionally, it costs limited tuning resources, due to instruction grouping strategy.

**State-of-The-Art Performance.** To the best of our knowledge, our method shows the best comprehensive continual tuning performance, especially significant enhancements in the performance of the LoRA Fine-tuning baseline (as shown in Figure 1).

## 2. Related Work

**Large Foundation Models.** LFMs own strong reasoning and complex contextual understanding capabilities, which can generate text sequences according to the input images and texts. With a frozen large language model and visual tower, BLIP2 (Li et al., 2023) bridged the gap between image and text modality through Qformer structure. Inspired by the notable reasoning power of instruction tuning

in LLMs, *e.g.,* LLaMA (Touvron et al., 2023a) and GPT (Floridi & Chiriatti, 2020; Achiam et al., 2023), LLaVA (Liu et al., 2024b) and MiniGPT4 (Zhu et al., 2023) adopted instruction tuning with only training a linear projection layer to align the image-text modalities. Recently, LLaVA-1.5 (Liu et al., 2024a) and MiniGPT5 (Zheng et al., 2023) refined the instructions and improved the performance across a wider range of multi-modality datasets.

**Continual Instruction Tuning Work:** (Zhang et al., 2023) first proposed the continual instruction tuning with LLMs. After that, TRACE, another continual instruction tuning benchmark, is designed to evaluate the general ability, instruction following, and safety for LLMs (Wang et al., 2023). Following them, (He et al., 2023) and (Chen et al., 2024a) designed distinct cross-modality continual instruction tuning benchmarks with InstructBLIP and LLaVA, respectively. Specifically, (He et al., 2023) adopted unique linear projection layers for each tuning dataset and utilized a key-query driven mechanism to infer the dataset identifier. (Chen et al., 2024a) employed the MOELoRA paradigm, assigning different LoRA weights to different datasets based on expert knowledge. Nevertheless, they still faced the issue of forgetting due to repeated training (Yang et al., 2023).

## 3. Preliminary

**Continual Instruction Tuning Definition:** CIT (Zhang et al., 2023) is defined to leverage LFMs to continually instruction-tune on new datasets without costly re-training. Compared with traditional continual learning, CIT differs in that it pays more attention to effectively leveraging the rich natural language instructions to prevent catastrophic forgetting and encourage knowledge transfer. Additionally, each task is an independent dataset. CIT can be described as a set of datasets $\mathcal{T}_{\text{seq}} = \{t_1, ..., t_T\}$ that arrives sequentially. Note that datasets in the stream can be any type and are not restricted to specific categories or domains. Each dataset $t_j \in \mathcal{T}_{\text{seq}}$ has a natural language instruction $I^{t_j}$, training set $\mathcal{D}_{\text{train}}^{t_j}$ and test set $\mathcal{D}_{test}^{t_j}$. The goal of CIT is to learn a single model $f$ from $\mathcal{T}_{\text{seq}}$ sequentially.

**Brief Review of The EMA Update Policy:** The EMA update has two kinds of parameters [1], one is parameters $\theta$ updating normally with gradient, and the other is EMA parameters $\theta^*$ updating as:

$$\theta_t^* = \beta_t \theta_{t-1}^* + (1 - \beta_t)\theta_t, \tag{1}$$

where $\beta_t$ is the EMA weight, $t$ and $t - 1$ is the training iteration. According to Eq.(1), performance on the current training iteration of $\theta_t^*$ is worse than $\theta_t$ because it only transfers a portion of the new knowledge from $\theta_t$ to $\theta_t^*$.

---

[1]Without a specific illustration, parameters in this paper refer to trainable parameters.

In Appendix A, we imply that $\theta^*$ is a weighted sum of $\theta_i(i \in \{1, t\})$, and the summation weight is a product of $\beta_i$ in different iterations. Each update can contribute to the EMA parameters by reviewing the previous parameters, which can make it have excellent stability. While for the traditional gradient update, the gradient only carries novel information, without reviewing the previous one. We further discuss that the performance of the EMA method is greatly affected by the summation weight. A stable EMA weight cannot be applied to flexible and various instruction datasets (see Appendix F). Thus, we are motivated to propose a dynamical update method for EMA weight.

## 4. Method

### 4.1. Proposition of Our Method

Primarily, we propose the following equations to simultaneously achieve the optimal ideal state with EMA update.

**Proposition 4.1.** *(Ideal State). Given an LFM with continual instruction tuning, with its parameters $\theta$ and EMA parameters $\theta^*$, after training on the iteration $t$, we can describe the ideal new knowledge transferring and the ideal old knowledge protecting as:*

$$\begin{cases} \mathcal{L}(\theta_t^*, x_t) = \mathcal{L}(\theta_t, x_t), \\ \qquad \theta_t^* = \theta_{t-1}^*. \end{cases} \tag{2}$$

The first equation of Eq.(2) represents ensuring the model performance on the new dataset with no change of training loss (**ideal new knowledge transferring**), inspired by the Optimal Brain Surgeon framework (Hassibi et al., 1993; LeCun et al., 1989; Frantar & Alistarh, 2022; Molchanov et al., 2022). The second equation of Eq.(2) represents preserving the model performance on old datasets with no change of model parameters (**ideal old knowledge protecting**).

***Discussion*:** From Proposition.4.1, we can further find that the starting point is model-independent. Thus our method can be extended to more LFMs, and parameter-efficient tuning paradigms, even more continual learning scenes.

### 4.2. Self-Adaption Dynamical Update Method

In order to find a dynamical update $\beta_t$, and realize Proposition.4.1, we start from a Taylor expansion of the loss function $\mathcal{L}$ around the individual parameter $\theta_t$ [2]. Our basis is that the gradient, which represents the discrepancy between the parameters and the new knowledge, is generated by the loss function.

$$\mathcal{L}(\theta) = \mathcal{L}(\theta_t) + \mathcal{L}'(\theta_t)(\theta - \theta_t) + \frac{\mathcal{L}''(\theta_t)}{2}(\theta - \theta_t)^2 + O(\theta - \theta_t)^3. \tag{3}$$

---

[2]For simplification, we omit the $x_t$ in $\mathcal{L}$

To further introduce $\theta_t^*$ in the Taylor expansion, we replace $\theta$ with $\theta_t^*$, and have:

$$\mathcal{L}(\theta_t^*) - \mathcal{L}(\theta_t) = \mathcal{L}'(\theta_t)(\theta_t^* - \theta_t) + \frac{\mathcal{L}''(\theta_t)}{2}(\theta_t^* - \theta_t)^2. \quad (4)$$

Notice that we have omitted the high-order infinitesimal term of $O(\theta - \theta_t)^3$. Additionally, we introduce the relaxation factor $\Delta\theta$ and have the stability constraint that $\theta_t^* = \theta_{t-1}^* + \Delta\theta$. Here, our intuition is from the perspective of EMA parameters update. The relaxation factor $\Delta\theta$ denotes the newly assimilated model parameters. Moving the left item to the right of the equation, we have:

$$\Delta\theta + \theta_{t-1}^* - \theta_t^* = 0. \quad (5)$$

Start from the stability constraint, combined with Eq.(1), we can obtain:

$$\theta_t^* - \theta_t = -\frac{\beta_t}{1 - \beta_t}\Delta\theta = \frac{\beta_t}{\beta_t - 1}\Delta\theta. \quad (6)$$

Please kindly refer to Appendix B for detailed demonstrations. In order to achieve the ideal new knowledge transferring and the ideal old knowledge protecting, we minimize the difference between $\mathcal{L}(\theta_t^*)$ and $\mathcal{L}(\theta_t)$, $\theta_t^*$ and $\theta_{t-1}^*$. Merging the two minimal situations, we have a unified optimal objective function and set up the following minimization problem:

$$\min \{\mathcal{L}(\theta_t^*) - \mathcal{L}(\theta_t) + \theta_t^* - \theta_{t-1}^*\}, \\ s.t. \Delta\theta + \theta_{t-1}^* - \theta_t^* = 0. \quad (7)$$

To further consider the constrained minimization problem, we use the method of Lagrange multipliers, which combines the objective function with the constraint by incorporating the Lagrange multiplier $\lambda$.

$$F = \mathcal{L}(\theta_t^*) - \mathcal{L}(\theta_t) + \theta_t^* - \theta_{t-1}^* + \lambda(\Delta\theta + \theta_{t-1}^* - \theta_t^*). \quad (8)$$

From Eq.(1), we can transfer the *s.t.* equation as:

$$\Delta\theta + \theta_{t-1}^* - \theta_t^* = \Delta\theta + \theta_{t-1}^* - [\beta_t\theta_{t-1}^* + (1 - \beta_t)\theta_t] \\ = \Delta\theta + (1 - \beta_t)(\theta_{t-1}^* - \theta_t). \quad (9)$$

After that, we substitute Eq.(56), Eq.(5), Eq.(6) and Eq.(9) into Eq.(57).

$$F = \mathcal{L}'(\theta_t)\frac{\beta_t}{\beta_t - 1}\Delta\theta + \frac{\mathcal{L}''(\theta_t)}{2}(\frac{\beta_t}{\beta_t - 1}\Delta\theta)^2 + \Delta\theta \\ + \lambda[\Delta\theta + (1 - \beta_t)(\theta_{t-1}^* - \theta_t)]. \quad (10)$$

Taking the derivative of the Lagrangian concerning $\beta_t$, we set it to zero and determine the direction in which the Lagrangian is stationary. This condition is essential for finding the optimal solution.

$$\frac{\partial F}{\partial \beta_t} = -\frac{1}{(\beta_t - 1)^2}\mathcal{L}'(\theta_t)\Delta\theta - \frac{\beta_t}{(\beta_t - 1)^3}\mathcal{L}''(\theta_t)\Delta\theta^2 \\ - \lambda(\theta_{t-1}^* - \theta_t) = 0. \quad (11)$$

By solving these equations, we obtain one feasible solution for $\beta_t$, which minimizes the objective function while satisfying the constraint:

$$\beta_t = \frac{\mathcal{L}'(\theta_t) + 1}{(\theta_t - \theta_{t-1}^*)\mathcal{L}''(\theta_t)}. \quad (12)$$

Please refer to Appendix C for the detailed deduction.

***Discussion***: Based on Eq.(12), we can discover that the optimal weight is basically related to the new gradient $\mathcal{L}'$ and the old parameters $\theta_{t-1}^*$, proving that the obtained EMA weight $\beta_t$ can make the trade-off between stability and plasticity. Detailed deduction about high-dimensional matrix $\Theta_t$ is presented in Appendix P.

### 4.3. Two Approximate Optimizations

In Eq.(12), the calculation of $\mathcal{L}''(\theta_t)$ involves the inverse of the Hessian matrix, which needs to obtain second-order partial derivatives. However, the above calculation is complex, leading to expensive memory and time-consuming, let alone for LLM, which further increases the training burden. Thus, how to approximately express the Hessian matrix without a complex calculation process becomes a challenge and urgently needs to be solved.

**Approximate Optimization Step I:** Considering that the Hessian matrix is obtained by partially deriving the gradients, we can approximate the derivative with the quotient of the differentiation. Here we recognize each iteration as a period of parameter update and further simplify the denominator. As a result, we have the following approximation equation to estimate the $\mathcal{L}''(\theta_t)$.

$$\mathcal{L}''(\theta_t) = \frac{\partial \mathcal{L}'(\theta_t)}{\partial \theta_t} = \frac{\mathcal{L}'(\theta_t) - \mathcal{L}'(\theta_{t-1})}{\theta_t - \theta_{t-1}}, \quad (13)$$

where the $\mathcal{L}'(\theta_t)$ represents gradients in the current iteration, and the $\mathcal{L}'(\theta_{t-1})$ denotes gradients in the last iteration.

$\theta_t$ in Eq.(12) represents the individual parameter, which causes the EMA weight $\beta_t$ to be individual parameter-wise. However, the training parameters $\theta$ always own a high dimension, leading to a huge computational cost. Thus, we propose to set the union parameter-wise $\beta_t$, *e.g.* one $\beta_t$ for one module layer, and utilize the following method to approximate the $\beta_t$.

**Approximate Optimization Step II:** Assuming that the EMA weight of each individual parameter in the same module layer would not change a lot. Here, we introduce L1-Norm and further approximate the $\beta_t$ as:

$$\beta_t \approx \|\frac{[\mathcal{L}'(\hat{\theta}_t) + 1](\hat{\theta}_t - \hat{\theta}_{t-1})}{(\hat{\theta}_t - \hat{\theta}_{t-1}^*)[\mathcal{L}'(\hat{\theta}_t) - \mathcal{L}'(\hat{\theta}_{t-1})]}\|. \quad (14)$$

$\hat{\theta}$ represents the whole layer parameters. If we have a coarse

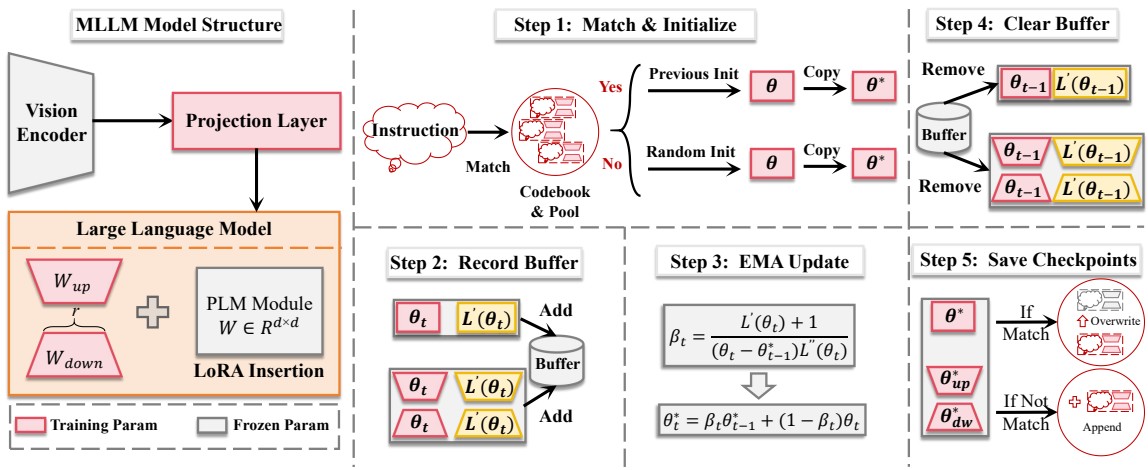

Figure 3. Overview of the proposed method (Taking MLLMs as an example). It is mainly divided into five steps. Step 1: Match with the codebook and initialize the training parameters. Step 2: Record the gradients and training parameters at task t. Step 3: Calculate the EMA weight and update the EMA parameters. Step 4: Clear the former records at task t-1. Step 5: Save the EMA parameters and instructions.

approximation that $\hat{\theta}_{t-1} = \hat{\theta}^*_{t-1}$, we can discover that:

$$\beta_t \approx \|\frac{\mathcal{L}'(\hat{\theta}_t) + 1}{\mathcal{L}'(\hat{\theta}_t) - \mathcal{L}'(\hat{\theta}_{t-1})}\| > 1. \quad (15)$$

Considering that the $\beta_t \in (0, 1)$, thus we adopt the $\beta_t = 0.99$ when $\beta_t$ exceeds the range of $(0, 1)$, where the constant value 0.99 is empirically obtained from experiments.

***Discussion***: The motivation for adopting L1-Norm to approximate the $\beta_t$ is L1-Norm occupies a few computation loads compared to other normalization methods. Detailed implementation of our method please refer to Appendix E.

### 4.4. Instruction Grouping Strategy

Inspired by L2P and Eproj (Wang et al., 2022b; He et al., 2023) [3], we design the following strategy to decide whether to retrain or expand the training parameters. We primarily build an instruction codebook and a training pool. Instructions are grouped in the codebook and each group is associated with a set of training parameters in the pool.

1. Before training a new task, we extract the instructions of the entire task and utilize the Term Frequency-Inverse Document Frequency (TF-IDF) model (Christian et al., 2016) to convert the instruction text into token vectors. Since the TF-IDF is a lightweight machine learning model, its computational cost is negligible (Detailed introduction can be found in Appendix H). 2. Based on the cosine similarity function, we calculate the similarities between the training instruction tokens and each token of saved instruction in the codebook. 3. If the maximum similarity is larger than the threshold $\epsilon$, we retrain the training parameters corresponding to the group which is the saved instruction of the

maximum similarity in. If not, we create a new set of training parameters. 4. After training, if we reuse the instruction, we overwrite the parameters checkpoint and append the unrepeated training instructions into the group. If not, we both restart a new parameters checkpoint and instruction group.

***Discussion***: We validate three kinds of instructions and the detailed grouping results can be found in Appendix I.

### 4.5. Overview of Our Method

Our method consists of five steps, as shown in Figure 3. Before training, the initial step is to match instructions of the current task with instructions in the codebook. If we find a similar instruction group, we utilize the corresponding parameters checkpoint to initialize parameters $\theta$ and EMA parameters $\theta^*$. Otherwise, we randomly initialize parameters $\theta$ and EMA parameters $\theta^*$. After the training of iteration $t$, gradients $\mathcal{L}'(\theta_t)$ and model parameters $\theta_t$ would be saved and involved in the EMA weight calculation process of the next iteration $t + 1$. Based on Eq.(14), we can obtain the adaptation EMA weight $\beta_t$. With the EMA weight $\beta_t$, we can update the EMA parameters from $\theta^*_{t-1}$ to $\theta^*_t$ based on Eq.(1). After the training of iteration $t$, to reduce the memory burden, we will clear the saved gradients $\mathcal{L}'(\theta_{t-1})$ and model parameters $\theta_{t-1}$. After training in each downstream dataset, we only save the EMA parameters $\theta^*$ and update the instruction codebook. For the detailed algorithm process please kindly refer to Appendix Q.

## 5. Experiments

### 5.1. Experimental Setup

**Implementation:** We adopt LLaVA-1.5/Qwen-VL (Liu et al., 2024b; Bai et al., 2023) as our backbone with inserted

---

[3] Detailed comparisons can be found in Appendix G

*Table 1.* Avg.ACC, Forgetting, and New.ACC performance comparisons between ours and baselines on LLaVA-7B.

| Method | Venue | Datasets | | | | | | | | Metrics | | |
|---|---|---|---|---|---|---|---|---|---|---|---|---|
| | | ScienceQA | TextVQA | ImageNet | GQA | VizWiz | Grounding | VQAv2 | OCR-VQA | Avg.ACC(↑) | Forgetting(↓) | New.ACC(↑) |
| Zero-shot | - | 49.91 | 2.88 | 0.33 | 2.08 | 0.90 | 0.00 | 0.68 | 0.17 | 7.12 | - | - |
| LoRA Fine-Tune(Hu et al., 2022) | ICLR'22 | 21.26 | 28.74 | 10.25 | 36.78 | 32.45 | 0.83 | 42.50 | 57.08 | 28.74 | 37.29 | 61.36 |
| MoELoRA(Chen et al., 2024a) | ArXiv'24 | 58.92 | 38.59 | 8.85 | 37.10 | 44.25 | 2.45 | 41.40 | 55.35 | 35.86 | 25.71 | 58.36 |
| LWF(Li & Hoiem, 2017) | TPAMI'16 | 63.14 | 39.60 | 8.90 | 34.83 | 14.53 | 2.48 | 40.67 | 62.35 | 33.31 | 22.32 | 52.58 |
| EWC(Kirkpatrick et al., 2017) | PNAS'17 | 67.41 | 40.41 | 8.18 | 35.05 | 37.88 | 2.67 | 41.27 | 61.02 | 36.74 | 20.51 | 54.68 |
| MT(Zhu et al., 2024) | ICML'24 | 79.63 | 55.47 | 35.64 | 58.70 | 44.37 | 32.20 | 62.21 | 61.59 | 53.73 | 14.03 | 66.00 |
| PGP(Qiao et al., 2024a) | ICLR'24 | 85.17 | 56.85 | 32.26 | 61.74 | 49.43 | 32.74 | 65.74 | 62.20 | 55.77 | 12.94 | 67.09 |
| EProj(He et al., 2023) | ArXiv'23 | 78.51 | 57.53 | 92.35 | 55.93 | 44.67 | 36.59 | 63.74 | 57.00 | 60.79 | 5.42 | 65.54 |
| **Ours** | - | **83.38** | **59.96** | **97.07** | **60.31** | **48.11** | **39.61** | **65.85** | **62.80** | **64.64** | **1.93** | **66.33** |
| Multi-Task | - | 56.77 | 49.35 | 95.55 | 56.65 | 53.90 | 30.09 | 59.50 | 55.65 | 57.18 | - | - |

*Table 2.* Avg.ACC, Forgetting, and New.ACC performance comparisons between ours and baselines on LLaVA-13B.

| Method | Venue | Datasets | | | | | | | | Metrics | | |
|---|---|---|---|---|---|---|---|---|---|---|---|---|
| | | ScienceQA | TextVQA | ImageNet | GQA | VizWiz | Grounding | VQAv2 | OCR-VQA | Avg.ACC(↑) | Forgetting(↓) | New.ACC(↑) |
| LoRA Fine-Tune(Hu et al., 2022) | ICLR'22 | 60.03 | 41.19 | 10.62 | 31.03 | 32.67 | 2.60 | 46.33 | 61.00 | 35.68 | 32.90 | 64.47 |
| MT(Zhu et al., 2024) | ICML'24 | 80.43 | 60.72 | 46.70 | 60.35 | 49.19 | 33.16 | 63.74 | 65.44 | 57.47 | 11.26 | 67.32 |
| PGP(Qiao et al., 2024a) | ICLR'24 | 82.50 | 60.64 | 49.15 | 62.53 | 49.43 | 37.37 | 65.57 | 65.82 | 59.13 | 10.11 | 67.98 |
| EProj(He et al., 2023) | ArXiv'23 | 77.65 | 58.93 | 92.31 | 60.22 | 38.27 | 33.77 | 64.39 | 65.80 | 61.42 | 5.84 | 66.53 |
| **Ours** | - | **83.94** | **61.40** | **97.05** | **62.61** | **43.99** | **39.72** | **66.29** | **65.78** | **65.10** | **2.31** | **67.12** |

LoRA (Hu et al., 2022) in the LLM side. For all methods, including ours and other baselines, during the continual instruction tuning, we freeze the vision encoder and LLM, with only training the projection layer and LoRA. We follow the datasets and tuning orders of the CoIN benchmark (Chen et al., 2024a), including ScienceQA (Lu et al., 2022), TextVQA (Singh et al., 2019), ImageNet (Deng et al., 2009), GQA (Hudson & Manning, 2019), VizWiz (Gurari et al., 2018), Grounding (Kazemzadeh et al., 2014; Mao et al., 2016), VQAv2 (Goyal et al., 2017), and OCR-VQA (Mishra et al., 2019). Experiments on continual instruction tuning with Large Language Models are shown in Appendix M.

**Compared Methods:** We compare our method against nine methods including (1) zero-shot and (2) LoRA Fine-Tune (Hu et al., 2022) (3) MoELoRA (Chen et al., 2024a); (4) LWF (Li & Hoiem, 2017); (5) EWC (Kirkpatrick et al., 2017); (6) EProj (He et al., 2023); (7) MT (Zhu et al., 2024); (8) PGP (Qiao et al., 2024a); (9) Multi-Task. The detailed descriptions of each method can be found in Appendix K.

**Evaluation Metrics:** We follow the most popular protocols for evaluation (Wang et al., 2022b;a; Smith et al., 2023; Qiao et al., 2024b), which are Average Accuracy (Simplified as Avg.Acc), Forgetting, and New Accuracy (Simplified as New.Acc). Please refer to Appendix L for more details.

## 5.2. Continual Instruction Tuning Results

**Comparison to SOTA:** Based on LLaVA-7B, we compare the performance in Table 1. We observe that our method can improve the best of other methods (EProj method) by +3.85@Avg.ACC, -3.49@Forgetting, and +0.79@New.ACC, demonstrating its excellent anti-forgetting and continual tuning ability. Although methods like LWF (Li & Hoiem, 2017) and EWC (Kirkpatrick et al., 2017) can resist forgetting, their plasticity is greatly influenced. PGP (Qiao et al., 2024a), MT (Zhu et al., 2024),

and EProj (He et al., 2023) can perform well both in stability and plasticity, while the Avg.ACC still needs to be improved. It is highlighted that our method owns the highest Avg.ACC, the lowest Forgetting and the comparatively higher New.ACC among these methods, which shows that our method can achieve the best trade-off between plasticity and stability.

Furthermore, based on LLaVA-13B, we compare the continual instruction tuning performance in Table 2 on larger MLLM. We observe that our method can improve the Avg.ACC (+3.68), the New.ACC (+0.59) and reduce the Forgetting (-3.53) compared with the best of other methods (EProj method), highlighting its superior anti-forgetting and continual tuning capabilities. When compared with other SOTA methods, *e.g.* PGP (Qiao et al., 2024a), MT (Zhu et al., 2024), and EProj (He et al., 2023), it is also noteworthy that our method owns the highest Avg.ACC, the lowest Forgetting and the comparatively higher New.ACC among these methods, which shows that our method can achieve the best trade-off between plasticity and stability in larger MLLM and lays the foundation for the application of our method to stronger MLLM with more parameters.

Additionally, based on Qwen-VL, we extend our evaluation to a wider range of MLLMs and compare the CIT performance in Table 3 for more comprehensive studies. The experimental results show that our method can also be effective for Qwen-VL, significantly improving its continual instruction tuning and anti-forgetting ability, which illustrates that our method owns generalization ability and can be applied to more MLLMs.

## 5.3. Robust Performance

To further validate the robustness of the proposed method, we adopt two robustness evaluation experiments with varied tuning task orders and distinct instruction types. (1). For the

*Table 3.* Avg.ACC, Forgetting, and New.ACC performance comparisons between ours and baselines on Qwen-VL.

| Method | Venue | Datasets | | | | | | | | Metrics | | |
|---|---|---|---|---|---|---|---|---|---|---|---|---|
| | | ScienceQA | TextVQA | ImageNet | GQA | VizWiz | Grounding | VQAv2 | OCR-VQA | Avg.ACC(↑) | Forgetting(↓) | New.ACC(↑) |
| LoRA Fine-Tune(Hu et al., 2022) | ICLR'22 | 31.05 | 42.45 | 29.57 | 55.57 | 15.30 | 40.33 | 67.75 | 47.80 | 41.23 | 19.36 | 58.17 |
| EWC (Kirkpatrick et al., 2017) | PNAS'17 | 64.30 | 58.67 | 44.04 | 57.73 | 38.16 | 48.04 | 66.98 | 41.76 | 52.46 | 8.68 | 50.67 |
| PGP(Qiao et al., 2024a) | ICLR'24 | 66.42 | 41.33 | 32.16 | 49.83 | 36.05 | 24.22 | 58.60 | 43.96 | 44.07 | 5.90 | 48.30 |
| **Ours** | - | **66.52** | **59.44** | **53.56** | **57.81** | **39.57** | **47.44** | **70.36** | **50.44** | **55.64** | **1.62** | **56.19** |

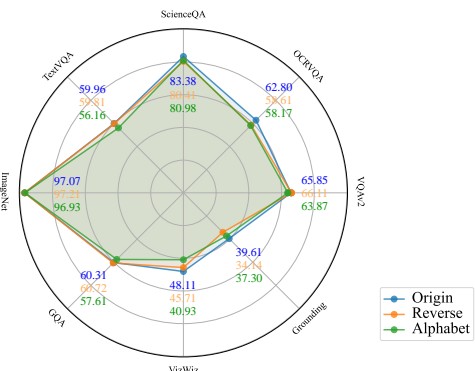

*Figure 4.* Radar chart of comparisons of final results on each task between varied tuning orders.

varied tuning task orders experiment, we set three different tuning orders (For detailed information please refer to Appendix N). The final results of each task are shown in Figure 4 and the evaluation metrics performance are shown in Table 4. We can find that although changing the order of tasks inevitably impacts the results of each task, the overall situation tends to stabilize with no significant fluctuations, as suggested by the similar distributions in Figure 4. In Table 4, because the knowledge acquired from previous tasks can either benefit or hinder subsequent training, New.ACC and Forgetting have a small range of fluctuations. Avg.ACC, as a comprehensive performance metric, its range of variation also exists caused by the changing of New.ACC and Forgetting. Notice here the fluctuations are affected by the intrinsic attribute of tuning sequence, which has no relationship with the specific method.

*Table 4.* Avg.ACC, Forgetting, and New.ACC performance comparisons between varied tuning orders.

| Order | Metrics | | |
|---|---|---|---|
| | Avg.ACC(↑) | Forgetting(↓) | New.ACC(↑) |
| Origin | 64.64 | 1.93 | 66.33 |
| Reverse | 62.84 | 2.25 | 64.81 |
| Alphabet | 61.49 | 2.75 | 63.90 |

*Table 5.* Avg.ACC, Forgetting, and New.ACC performance comparisons between distinct training instructions.

| Type | Metrics | | |
|---|---|---|---|
| | Avg.ACC(↑) | Forgetting(↓) | New.ACC(↑) |
| Origin | 64.64 | 1.93 | 66.33 |
| Diverse | 64.36 | 0.45 | 64.76 |
| 10Type | 62.92 | 2.86 | 65.41 |

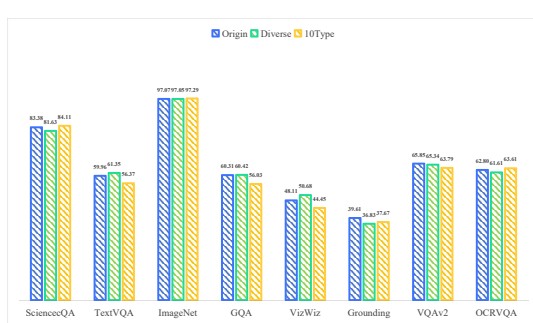

*Figure 5.* Column chart of comparisons of final results on each task between distinct instructions.

(2) For distinct instruction types experiment, we employ three types of instructions (Detailed information please refer to Appendix O). The evaluation metrics performance are shown in Table 5. We can find that although New.ACC and Forgetting have a small range of fluctuations, but for Avg.ACC, as a comprehensive performance metric, its range of variation is almost invisible. We further research the Final Accuracy of each dataset in various types of instructions, and the results are shown in Figure 5. The Final Accuracy of the same dataset in each type looks very close to each other, which also proves the robustness of our method.

Based on the above observations, we conclude that the fluctuations in New.ACC and Forgetting are caused by changes in instruction type and dataset tuning order. However, our method has strong robustness, which can maintain the Avg.ACC at a stable level in each type of training strategy.

### 5.4. Analysis of Examples

In Figure 6, we show three testing instances with outputs from distinct methods and observe that our method can revise errors caused by continual instruction tuning. For the GQA instance (as shown in the left part), our method can keep the geographic location recognition capability of MLLMs after tuning. LoRA Fine-Tune seems to lose such ability and mistakes the chair for table. Although MT and EProj can recognize the object, their outputs are still not aligned with the ground truth, *i.e.* the first letter is not capitalized and the plural form is used. For the ImageNet instance (as shown in the middle part), our method can preserve the fine-grained knowledge of MLLMs after tuning. In contrast, LoRA Fine-Tune can only recognize the coarse attribute of the object. MT even appears the hallucination. For the Grounding instance (as shown in the right part),

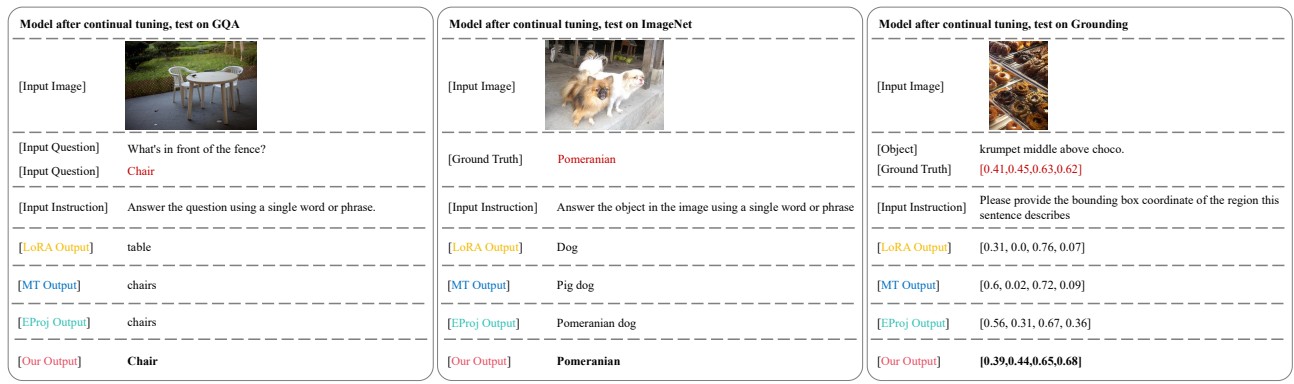

*Figure 6.* Visualization of multimodal continual instruction tuning examples, comparison between LoRA, MT, EProj, and ours.

*Table 6.* Ablation study results for each proposed component.

| Method | Metrics | | |
|---|---|---|---|
| | Avg.ACC(↑) | Forgetting(↓) | New.ACC(↑) |
| LoRA Baseline | 28.74 | 37.29 | 61.36 |
| Stable EMA ($\beta = 0.99$) | 48.09 | 16.24 | 62.30 |
| Dynamical EMA | 55.33 | 7.04 | 61.49 |
| Dynamical EMA + Instruction Grouping | **64.64** | **1.93** | **66.33** |

our method can protect the visual grounding ability in real scenarios. It is obvious that the IoU between our output and ground truth is the largest compared to other methods. Additionally, we are surprised to find that our method can spontaneously suppress the occurrence of hallucinations in the continual instruction tuning. (Zhai et al., 2023) deem that the hallucination in large models is related to the catastrophic forgetting in continual tuning. The above view is consistent with our multiple rounds of dialogue results, which are shown in Appendix D.

**5.5. Ablation Study**

To validate the efficiency of each component in the proposed method, starting from the LoRA Fine-Tune baseline, we incrementally add the component and compare the continual instruction tuning performances. Results are shown in Table 6. The experimental results demonstrate that each proposed component is efficient in enhancing accuracy and reducing forgetting of MLLMs.

In order to demonstrate the effectiveness of the proposed dynamic EMA update method, we compare it with the fixed EMA weight method, where the fixed EMA weight is set to 0.99, as suggested by the experiments. We can observe that, compared to the fixed EMA weight method, the dynamic EMA update method significantly improves the anti-forgetting ability (-9.20@Forgetting) and continual instruction tuning ability (+7.24@Avg.ACC). Moreover, with the aid of the instruction grouping strategy, both forgetting resistance (-5.11@Forgetting) and continual instruction tuning ability (+9.31@Avg.ACC) show further significant

improvements. In summary, compared to the original LoRA Fine-Tune method, our method greatly enhances the metrics of model in Avg.ACC (+35.90), New.ACC (+4.97), and reduces Forgetting (-35.36), achieving an optimal balance between plasticity and stability.

**6. Conclusion**

To enable LFMs to possess the ability of continual instruction tuning and further resist forgetting, we propose a general continual instruction tuning framework. Combined with the exponential moving average, the proposed method can protect previous knowledge and incorporate new knowledge at the same time. By solving a set of equations based on the Lagrange multiplier method, we obtain the self-adaption weight of EMA in each update process. Subsequently, two compensation mechanisms are further introduced to alleviate the computational costs. Additionally, based on the instruction grouping strategy, we can retrain the parameters of semantic-similar instructions and limitedly expand the parameters of semantic-irrelevant instructions. In the testing phase, we also utilize the strategy to match the most suitable parameters for the instances. Experiments on MLLMs and LLMs show that our approach not only owns excellent anti-forgetting but also well continual tuning performance. Due to computational resource constraints, our current focus is primarily on the image and text continual instruction tuning. In the future, we aim to extend our method to continual instruction tuning benchmarks with more modalities and more continual tuning scenarios.

## Acknowledgements

This work is supported by the National Natural Science Foundation of China (No.62222602, No.62176092, No.U23A20343, No.62476090), Natural Science Foundation of Shanghai (23ZR1420400), Shanghai Sailing Program (23YF1410500), CCF-Tencent RAGR20240122, Young Elite Scientists Sponsorship Program by CAST (YESS20240780).

## Impact Statement

1.The potential of our method to enhance LFMs' continual instruction tuning performance and mitigate catastrophic forgetting. 2.Its relevance to real-world applications, such as lifelong AI assistants and LLM continual evolution. 3.The datasets utilized in this paper are derived from publicly accessible sources. Rigorous data processing procedures have been implemented to ensure the exclusion of any personally sensitive information, in full compliance with relevant privacy protection and ethical standards.

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

## A. Decomposition of EMA Update

In the EMA update, it exists two kinds of parameters, normally parameters $\theta$ and EMA parameters $\theta^*$. At iteration 1, $\theta_1^*$ is updated according to Eq.(1) as:

$$\theta_1^* = \beta_1\theta_0^* + (1 - \beta_1)\theta_1. \tag{16}$$

Then at iteration 2, by replacing Eq.(16), $\theta_2^*$ is updated as:

$$\begin{aligned}
\theta_2^* = \beta_2\theta_1^* + (1 - \beta_2)\theta_2 &= \beta_2[\beta_1\theta_0^* + (1 - \beta_1)\theta_1] + (1 - \beta_2)\theta_2 \\
&= \beta_2\beta_1\theta_0^* + \beta_2(1 - \beta_1)\theta_1 + (1 - \beta_2)\theta_2.
\end{aligned} \tag{17}$$

After that, at iteration 3, by replacing Eq.(17), $\theta_3^*$ is updated as:

$$\begin{aligned}
\theta_3^* = \beta_3\theta_2^* + (1 - \beta_3)\theta_3 &= \beta_3[\beta_2\beta_1\theta_0^* + \beta_2(1 - \beta_1)\theta_1 + (1 - \beta_2)\theta_2] + (1 - \beta_3)\theta_3 \\
&= \beta_3\beta_2\beta_1\theta_0^* + \beta_3\beta_2(1 - \beta_1)\theta_1 + \beta_3(1 - \beta_2)\theta_2 + (1 - \beta_3)\theta_3.
\end{aligned} \tag{18}$$

Observing the equation form, based on the method of summarization and induction, we have the following assumption for iteration $n - 1$:

$$\theta_{n-1}^* = \prod_{i=1}^{n-1} \beta_i \cdot \theta_0^* + \sum_{i=1}^{n-1} (1 - \beta_i) \cdot \prod_{j=i+1}^{n-1} \beta_j \cdot \theta_i. \tag{19}$$

Finally, at iteration $n$, by replacing Eq.(19), $\theta_n^*$ is updated as:

$$\begin{aligned}
\theta_n^* &= \beta_n[\prod_{i=1}^{n-1} \beta_i \cdot \theta_0^* + \sum_{i=1}^{n-1} (1 - \beta_i) \cdot \prod_{j=i+1}^{n-1} \beta_j \cdot \theta_i] + (1 - \beta_n)\theta_n \\
&= \prod_{i=1}^{n} \beta_i \cdot \theta_0^* + \sum_{i=1}^{n-1} (1 - \beta_i) \cdot \prod_{j=i+1}^{n} \beta_j \cdot \theta_i + (1 - \beta_n)\theta_n \\
&= \prod_{i=1}^{n} \beta_i \cdot \theta_0^* + \sum_{i=1}^{n} (1 - \beta_i) \cdot \prod_{j=i+1}^{n} \beta_j \cdot \theta_i.
\end{aligned} \tag{20}$$

It can be found that Eq.(20) also has the same form as Eq.(19), which means that the assumption is established. Due to utilizing $\theta_0$ to initialize $\theta_0^*$, EMA parameters $\theta_t^*$ can be represented by normally parameter $\theta$ as:

$$\theta_t^* = \prod_{i=1}^{t} \beta_i \cdot \theta_0 + \sum_{i=1}^{t} (1 - \beta_i) \cdot \prod_{j=i+1}^{t} \beta_j \cdot \theta_i. \tag{21}$$

## B. Proof of relationship between $\theta_t$, $\theta_t^*$ and $\Delta\theta$

From *s.t.* constraint, we have:

$$\Delta\theta = \theta_t^* - \theta_{t-1}^*, \tag{22}$$

$$\theta_{t-1}^* = \theta_t^* - \Delta\theta. \tag{23}$$

Replace $\theta_{t-1}^*$ with $\theta_t^* - \Delta\theta$ in Eq.(1):

$$\theta_t^* = \beta_t(\theta_t^* - \Delta\theta) + (1 - \beta_t)\theta_t. \tag{24}$$

Rearrange the above equation and have:

$$\theta_t^* - \theta_t = \beta_t(\theta_t^* - \theta_t) - \beta_t\Delta\theta, \tag{25}$$

$$(1 - \beta_t)(\theta_t^* - \theta_t) = -\beta_t\Delta\theta. \tag{26}$$

Finally, we can achieve that:

$$\theta_t^* - \theta_t = -\frac{\beta_t}{1 - \beta_t}\Delta\theta = \frac{\beta_t}{\beta_t - 1}\Delta\theta. \tag{27}$$

## C. $\beta_t$ **Solving Process**

With introducing Eq.(23) and Eq.(6), we can represent $\theta_{t-1}^* - \theta_t$ as:

$$\theta_{t-1}^* - \theta_t = \theta_t^* - \Delta\theta - \theta_t = \frac{\beta_t}{\beta_t - 1}\Delta\theta - \Delta\theta = \frac{\Delta\theta}{\beta_t - 1}. \tag{28}$$

Taking the derivative of the Lagrangian to $\Delta\theta$ and setting it to zero as Eq.(11), we have:

$$\frac{\partial F}{\partial \Delta\theta} = \frac{\beta}{(\beta - 1)}\mathcal{L}'(\theta_t) + \frac{\beta^2}{(\beta - 1)^2}\mathcal{L}''(\theta_t)\Delta\theta + 1 + \lambda = 0. \tag{29}$$

Further, we substitute Eq.(28) and Eq.(29) into Eq.(11), and have:

$$0 = -\frac{1}{(\beta_t - 1)^2}\mathcal{L}'(\theta_t)\Delta\theta - \frac{\beta_t}{(\beta_t - 1)^3}\mathcal{L}''(\theta_t)\Delta\theta^2 - [-\frac{\beta_t}{(\beta_t - 1)}\mathcal{L}'(\theta_t) -$$
$$\frac{\beta_t^2}{(\beta_t - 1)^2}\mathcal{L}''(\theta_t)\Delta\theta - 1](\theta_{t-1}^* - \theta_t), \tag{30}$$

$$0 = -\frac{1}{(\beta_t - 1)^2}\mathcal{L}'(\theta_t)\Delta\theta - \frac{\beta_t}{(\beta_t - 1)^3}\mathcal{L}''(\theta_t)\Delta\theta^2 + \frac{\beta_t}{(\beta_t - 1)^2}\mathcal{L}'(\theta_t)\Delta\theta +$$
$$\frac{\beta_t^2}{(\beta_t - 1)^3}\mathcal{L}''(\theta_t)\Delta\theta^2 + \frac{\Delta\theta}{\beta_t - 1}, \tag{31}$$

$$0 = \frac{-1 + \beta_t}{(\beta_t - 1)^2}\mathcal{L}'(\theta_t)\Delta\theta + \frac{-\beta_t + \beta_t^2}{(\beta_t - 1)^3}\mathcal{L}''(\theta_t)\Delta\theta^2 + \frac{\Delta\theta}{\beta_t - 1}, \tag{32}$$

$$0 = \frac{1}{(\beta_t - 1)}\mathcal{L}'(\theta_t)\Delta\theta + \frac{\beta_t}{(\beta_t - 1)^2}\mathcal{L}''(\theta_t)\Delta\theta^2 + \frac{\Delta\theta}{\beta_t - 1}, \tag{33}$$

$$0 = \Delta\theta[\frac{1}{(\beta_t - 1)}\mathcal{L}'(\theta_t) + \frac{\beta_t}{(\beta_t - 1)^2}\mathcal{L}''(\theta_t)\Delta\theta + \frac{1}{\beta_t - 1}]. \tag{34}$$

By observation, we can find one solution that $\Delta\theta = 0$, which means that $\theta_t^* = \theta_{t-1}^*$ and $\beta_t = 1$. Obviously, it is not the global optimal solution due to the lack of updates to EMA parameters.

Then, we can find another solution through the following equation:

$$0 = \frac{1}{(\beta_t - 1)}\mathcal{L}'(\theta_t) + \frac{\beta_t}{(\beta_t - 1)^2}\mathcal{L}''(\theta_t)\Delta\theta + \frac{1}{\beta_t - 1}. \tag{35}$$

Due to the situation that $\beta_t - 1 = 0$ has been discussed, we can remove it unlimited:

$$0 = \mathcal{L}'(\theta_t) + \frac{\beta_t}{(\beta_t - 1)}\mathcal{L}''(\theta_t)\Delta\theta + 1. \tag{36}$$

From Eq.(28), we can get:

$$\Delta\theta = (\theta_{t-1}^* - \theta_t)(\beta_t - 1). \tag{37}$$

Substitute Eq.(37) into Eq.(36):

$$0 = \mathcal{L}'(\theta_t) + \beta_t(\theta_{t-1}^* - \theta_t)\mathcal{L}''(\theta_t) + 1. \tag{38}$$

Finally, we obtain another solution for $\beta_t$ that:

$$\beta_t = \frac{\mathcal{L}'(\theta_t) + 1}{(\theta_t - \theta_{t-1}^*)\mathcal{L}''(\theta_t)}. \tag{39}$$

**C.1. Discussions of $\beta_t$**

**Satisfy the *s.t.* equation**

According to Eq.(28), we have already proved:

$$(\theta^*_{t-1} - \theta_t) = \frac{\Delta\theta}{\beta_t - 1}, \tag{40}$$

$$(\theta^*_{t-1} - \theta_t)(\beta_t - 1) = \Delta\theta. \tag{41}$$

Thus, we can achieve the *i.e.* constraint with the solution as:

$$\begin{aligned}
\Delta\theta + \theta^*_{t-1} - \theta^*_t &= (\theta^*_{t-1} - \theta_t)(\beta_t - 1) + \theta^*_{t-1} - \theta^*_t = (\theta^*_{t-1} - \theta_t)\beta_t + \theta_t - \theta^*_t \\
&= (\theta^*_{t-1} - \theta_t)\beta_t + \theta_t - [\beta_t\theta^*_{t-1} + (1 - \beta_t)\theta_t] \\
&= \theta^*_{t-1}\beta_t - \theta_t\beta_t + \theta_t - \beta_t\theta^*_{t-1} - \theta_t + \beta_t\theta_t = 0.
\end{aligned} \tag{42}$$

## D. Cases of Multiple Rounds of Dialogue

In this section, we test the zero-shot performance of MLLMs continually fine-tuned with our method on multiple rounds of dialogue tasks. Images and questions are from (Liu et al., 2024b). To have a comparison, we also test the zero-shot performance of MLLMs continually fine-tuned with the baseline on multiple rounds of dialogue tasks.

---

**Visual input example, Funny Image:**

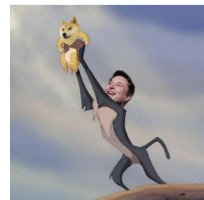

---

**llava-lora (baseline)**

**Human:** What is the name of the man in the photo?
**Assistant:** the man in the photo is named elon musk
**Human:** Where is the interesting aspect of this picture?
**Assistant:** the interesting aspect of this picture is the man holding a dog
**Human:** What is Elon Musk holding in his hand?
**Assistant:** elon musk is holding a dog
**Human:** What kind of dog is in elon's hand?
**Assistant:** a puppy

---

**llava-dema (ours)**

**Human:** What is the name of the man in the photo?
**Assistant:** Elon Musk
**Human:** Where is the interesting aspect of this picture?
**Assistant:** Elon Musk's face
**Human:** What is Elon Musk holding in his hand?
**Assistant:** puppy
**Human:** What kind of puppy is in Elon's hand?
**Assistant:** Shiba Inu

---

*Figure 7.* Case I of multiple rounds of dialogue, red marks the rough and low-quality answers generated by the baseline, and blue marks that our method can generate fine-grained and high-quality answers. We believe that this skill is learned from the downstream datasets. Besides that, we also discover that generated answers of our method are more concise and to the point, directly addressing the problem.

**Visual input example, Extreme Ironing:**

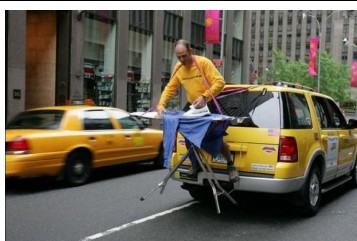

---

### llava-lora (baseline)

**Human:** Can you describe the image?
**Assistant:** In the image, a man is standing on the back of a yellow truck, **holding a clothesline** with clothes hanging on it. He appears to be **washing clothes** in the middle of a busy street. The truck is driving down the road, and there are other vehicles around, including a taxi and a car**.**

There are also **several traffic lights visible in the scene**, indicating that the area is likely a busy intersection. The man is focused on his task, despite the bustling environment around him.

**Human:** Do you think there is anything that goes against human intuition?
**Assistant:** Yes, the image of a man **washing clothes** in the middle of a busy street goes against human intuition. Typically, people would expect to see someone **washing clothes** in a more private or secluded area, such as a backyard or a laundromat. However, in this case, the man has chosen to **wash clothes** in the middle of a busy street, which is an unusual and unconventional location for such an activity.

---

### llava-dema (ours)

**Human:** Can you describe the image?
**Assistant:** In the image, a man is standing on a folding chair in the back of a yellow taxi cab, **ironing clothes**. The taxi is driving down a busy city street, surrounded by other vehicles such as a car and a truck. The man appears to be focused on his task, possibly trying to **dry his clothes quickly** in the moving vehicle.

**Human:** Do you think there is anything that goes against human intuition?
**Assistant:** Yes, the image of a man **ironing clothes** in the back of a taxi cab is unusual and goes against human intuition. It is not a typical scenario one would expect to see, as ironing clothes is usually done in a more comfortable and controlled environment, such as a home or a laundry room. The man's decision to **iron clothes** in the back of a taxi cab suggests that he is **trying to save time or that he might be in a hurry**.

---

*Figure 8.* Case II of multiple rounds of dialogue, red marks the hallucinations existing in the sentences generated by the baseline, and blue marks that our method can overcome the occurrence of hallucinations in relevant statements and generate correct answers. Besides that, we also discover that our method can provide reasonable explanations for some unusual behaviors.

---

**Visual input example, Cooking Recipe:**

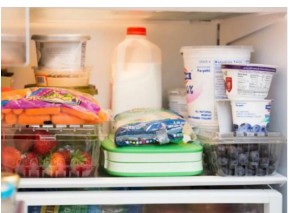

---

**llava-lora (baseline)**

**Human:** What are the meals that I can cook with these?
**Assistant:** With the items in the refrigerator, you can prepare a variety of meals. For example, you can make a fruit salad using the strawberries and blueberries. You can also use the carrots to prepare a healthy vegetable stir-fry, a soup, or a salad. The milk can be used to make a smoothie, a sauce, or a creamy soup. The eggs can be used to make a fried egg, an omelette, or a baked dish. Additionally, you can use the cheese to make a pizza, a grilled cheese sandwich, or a cheese-based sauce. The yogurt can be used to make a yogurt parfait, a dip, or a sauce. The butter can be used to cook or bake various dishes, such as bread or pastries. The ice cream can be used to make a dessert or a topping for other dishes.
**Human:** I would like to do the fruit salad. Show me the recipe.
**Assistant:** Sure, here's a simple recipe for a fruit salad:
Ingredients:
* 1 cup strawberries, hulled and sliced
* 1 cup blueberries, rinsed and drained
* 1/2 cup **raspberries**, rinsed and drained
* 1/2 cup **grapes**, seedless
* 1/2 cup **cantaloupe**, diced
* 1/4 cup honey or sugar (optional)
* 1/4 cup lemon juice (optional)
* 1/4 cup chopped fresh mint leaves (optional)
Instructions:
1. In a large bowl, combine the strawberries, blueberries, raspberries, grapes, and cantaloupe.
2. If desired, add honey or sugar to taste.
3. If desired, add lemon juice to the bowl for a tangy flavor.
4. Gently mix the fruit together.
5. Garnish with chopped fresh mint leaves, if desired.
Enjoy your fruit salad!

---

**llava-dema (ours)**

**Human:** What are the meals that I can cook with these?
**Assistant:** With the variety of fruits, vegetables, and milk available in the refrigerator, you can prepare a range of healthy and delicious meals. Some ideas include:
**1. Fruit salad:** Combine the strawberries, blueberries, and any other fruits available in the refrigerator. You can add a squeeze of lemon juice and a sprinkle of sugar for a simple, refreshing dessert.
**2. Vegetable stir-fry:** Chop up the carrots and any other vegetables available in the refrigerator. Heat some oil in a pan and stir-fry the vegetables with your choice of protein (e.g., tofu, chicken, or beef) and a flavorful sauce.
**3. Smoothie:** Blend the milk with the blueberries, strawberries, and any other fruits available in the refrigerator. You can also add some ice and a scoop of protein powder for a nutritious and delicious smoothie.
**4. Vegetable soup:** Use the carrots and any other vegetables available in the refrigerator to make a vegetable soup. You can add some milk or a vegetable broth for added flavor and nutrition.
**5. Omelette:** Whisk eggs with milk and season with salt and pepper. Cook the omelette in a pan and add the carrots and any other vegetables available in the refrigerator for a nutritious and filling breakfast.
These are just a few examples of the many meals you can prepare using the ingredients available in the refrigerator.
**Human:** I would like to do the fruit salad. Show me the recipe.
**Assistant:** Sure, here's a simple recipe for a fruit salad using the ingredients available in the refrigerator:
Ingredients:
* Strawberries
* Blueberries
* Carrots (optional)
* Milk
* Lemon juice (optional)
* Sugar (optional)
Instructions:
1. Wash the strawberries and blueberries thoroughly.
2. Chop the strawberries into bite-sized pieces.
3. Wash the blueberries and chop them into bite-sized pieces.
4. If you'd like to add carrots, wash and chop them as well.
5. In a large bowl, combine the chopped strawberries, blueberries, and carrots (if using).
6. Pour the milk over the fruit mixture and stir gently.
7. If you'd like to add a little sweetness, squeeze a lemon wedge over the fruit mixture and sprinkle with sugar.
8. Serve the fruit salad immediately or refrigerate for a few hours to allow the flavors to meld together.
Enjoy your delicious and refreshing fruit salad!

---

*Figure 9.* Case III of multiple rounds of dialogue, red marks the hallucinations existing in the sentences generated by the baseline, which have not existed in the image. Blue marks that our method can generate answers that are clearer and more aligned with human reading habits.

---

**Visual input example, Webpage Making:**

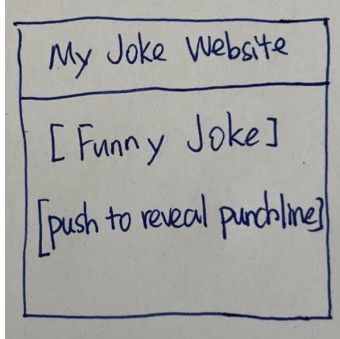

---

**llava-lora (baseline)**

**Human:** Please turn this mock-up into an interactive website using html/js, and replace the joke with a real joke.
**Assistant:** I'm sorry, but I am not able to create an interactive website using html/js. However, I can provide you with a real joke:

Why was the math book sad?

Because it had too many problems.

---

**llava-dema (ours)**

**Human:** Please turn this mock-up into an interactive website using html/js, and replace the joke with a real joke.
**Assistant:** Sure, I can help you with that! Here's a simple HTML/JS mock-up of the joke website:
```html
<!DOCTYPE html>
<html>
 <head>
  <title>My Joke Website</title>
 </head>
 <body>
  <h1>My Joke Website</h1>
  <h2>Funny Joke</h2>
  <p>Push to reveal punchline</p>
  <button onclick="revealPunchline()">Reveal Punchline</button>
  <p id="punchline"></p>
  <script>
   function revealPunchline() {
     document.getElementById("punchline").innerHTML = "Why did the tomato turn red?";
   }
  </script>
 </body>
</html>
```
This is a simple mock-up of the joke website, which includes a button to reveal the punchline. The punchline is stored in the `punchline` element, which is hidden by default. When the button is clicked, the `revealPunchline()` function is called, which sets the text of the `punchline` element to "Why did the tomato turn red?".

---

*Figure 10.* Case IV of multiple rounds of dialogue, after continually fine-tuning, the baseline has forgotten the pre-trained knowledge and lost the ability to write HTML/JS code. While our method can protect the previous knowledge and still retain the ability of webpage code writing after continually fine-tuning.

# E. Detailed Implementation

Based on Eq.(14), we continue to further simplify it as:

$$
\begin{aligned}
\beta_t &\approx \left\| \frac{[\mathcal{L}'(\hat{\theta}_t) + 1](\hat{\theta}_t - \hat{\theta}_{t-1})}{(\hat{\theta}_t - \hat{\theta}^*_{t-1})[\mathcal{L}'(\hat{\theta}_t) - \mathcal{L}'(\hat{\theta}_{t-1})]} \right\| \\
&= \left\| \frac{[\mathcal{L}'(\hat{\theta}_t) + 1](\hat{\theta}_t - \hat{\theta}^*_{t-1} + \hat{\theta}^*_{t-1} - \hat{\theta}_{t-1})}{(\hat{\theta}_t - \hat{\theta}^*_{t-1})[\mathcal{L}'(\hat{\theta}_t) - \mathcal{L}'(\hat{\theta}_{t-1})]} \right\| \\
&= \left\| \frac{\mathcal{L}'(\hat{\theta}_t) + 1}{\mathcal{L}'(\hat{\theta}_t) - \mathcal{L}'(\hat{\theta}_{t-1})} - \frac{[\hat{\theta}_{t-1} - \hat{\theta}^*_{t-1}][\mathcal{L}'(\hat{\theta}_t) + 1]}{(\hat{\theta}_t - \hat{\theta}^*_{t-1})[\mathcal{L}'(\hat{\theta}_t) - \mathcal{L}'(\hat{\theta}_{t-1})]} \right\| \\
&= \left\| \frac{\mathcal{L}'(\hat{\theta}_t) + 1 - \mathcal{L}'(\hat{\theta}_{t-1}) + \mathcal{L}'(\hat{\theta}_{t-1})}{\mathcal{L}'(\hat{\theta}_t) - \mathcal{L}'(\hat{\theta}_{t-1})} - \frac{[\hat{\theta}_{t-1} - \hat{\theta}^*_{t-1}][\mathcal{L}'(\hat{\theta}_t) + 1]}{(\hat{\theta}_t - \hat{\theta}^*_{t-1})[\mathcal{L}'(\hat{\theta}_t) - \mathcal{L}'(\hat{\theta}_{t-1})]} \right\| \\
&= \left\| 1 + \frac{1 + \mathcal{L}'(\hat{\theta}_{t-1})}{\mathcal{L}'(\hat{\theta}_t) - \mathcal{L}'(\hat{\theta}_{t-1})} - \frac{[\hat{\theta}_{t-1} - \hat{\theta}^*_{t-1}][\mathcal{L}'(\hat{\theta}_t) + 1]}{(\hat{\theta}_t - \hat{\theta}^*_{t-1})[\mathcal{L}'(\hat{\theta}_t) - \mathcal{L}'(\hat{\theta}_{t-1})]} \right\|.
\end{aligned}
\tag{43}
$$

Additionally, by observation in experiments, we find that $\|\mathcal{L}'(\hat{\theta}_{t-1}) + 1\| \ll \|\mathcal{L}'(\hat{\theta}_t) - \mathcal{L}'(\hat{\theta}_{t-1})\|$, leading to:

$$
\left\| \frac{1 + \mathcal{L}'(\hat{\theta}_{t-1})}{\mathcal{L}'(\hat{\theta}_t) - \mathcal{L}'(\hat{\theta}_{t-1})} \right\| \approx 0.
\tag{44}
$$

Therefore, Eq.(43) could be transferred as:

$$
\beta_t \approx \left\| 1 - \frac{[\hat{\theta}_{t-1} - \hat{\theta}^*_{t-1}][\mathcal{L}'(\hat{\theta}_t) + 1]}{(\hat{\theta}_t - \hat{\theta}^*_{t-1})[\mathcal{L}'(\hat{\theta}_t) - \mathcal{L}'(\hat{\theta}_{t-1})]} \right\| \approx 1 - \left\| \frac{[\hat{\theta}_{t-1} - \hat{\theta}^*_{t-1}][\mathcal{L}'(\hat{\theta}_t) + 1]}{(\hat{\theta}_t - \hat{\theta}^*_{t-1})[\mathcal{L}'(\hat{\theta}_t) - \mathcal{L}'(\hat{\theta}_{t-1})]} \right\|.
\tag{45}
$$

The above is our final result, and we approximate $\beta_t$ using Eq.(45) in implementation.

# F. Influence of EMA summation weight

We conduct a toy experiment to discuss that distinct stable EMA weights can greatly influence the continual instruction tuning results. We fine-tune on six tasks, including ScienceQA, TextVQA, GQA, VizWiz, VQAv2, and OCRVQA, based on the LLaVA-7B backbone. To have a comparison, we utilize three EMA weights and the results are shown in Table 7.

Table 7. Continual Instruction Tuning Performance of Distinct EMA Weights

| EMA Weight | ScienceQA | TextVQA | GQA | VizWiz | VQAv2 | OCRVQA | Avg.ACC | Forgetting | New.ACC |
|---|---|---|---|---|---|---|---|---|---|
| $\beta = 0.990$ | 76.37 | 62.13 | 62.20 | 58.60 | 67.24 | 63.91 | 60.76 | 5.17 | 65.08 |
| | 73.69 | 54.67 | 59.60 | 47.26 | 65.45 | 63.91 | | | |
| $\beta = 0.991$ | 76.28 | 62.23 | 61.07 | 58.00 | 67.20 | 63.48 | 58.83 | 7.06 | 64.71 |
| | 72.88 | 52.29 | 58.18 | 41.84 | 64.29 | 63.48 | | | |
| $\beta = 0.992$ | 76.47 | 61.94 | 61.91 | 58.09 | 67.46 | 65.54 | 60.69 | 5.46 | 65.24 |
| | 72.88 | 55.40 | 59.44 | 45.40 | 65.46 | 65.54 | | | |

From Table 7, we further discover that the performance of the EMA method is greatly affected by the summation weight, and the results of continual instruction tuning obtained under different EMA weights vary significantly.

# G. Comparisons Between Our Method With Others

Notice that existing methods *e.g.* L2P (Wang et al., 2022b) and EProj (He et al., 2023) also own the training parameters (Prompt or Linear Projection Layer) selection mechanism, which are similar to our instruction grouping and parameters expansion strategy. Thus, in this section, we analyze and illustrate the differences between our method and theirs.

L2P owns a fixed prompt pool, which consists of key-prompt pairs. During the training and testing stage, L2P adopts a frozen pre-trained Vision-Transformer to extract the image embedding and match it with the keys by using cosine similarity. After matching, it would find the prompts with the top-K similarity. Notice here, besides the training cost of prompt parameters, key parameters also need to be optimized in the training iteration.

EProj proposes a novel task-similarity model expansion, which decides whether to retrain or expand the image-to-text projection layer by measuring the cosine similarity between the current task and history tasks. To be specific, it would collect the mean embeddings of image $e(v)$, instruction $e(t)$, and output $e(o)$ of the entire dataset to comprise the task embeddings. The corresponding embeddings are encoded by the BERT model and the frozen Vision-Transformer in MLLMs respectively. During the training stage, it also extra trains key parameters to match with the input task embeddings aiming to retrieve the corresponding projection layer at the testing time.

While for our method, (1). It does not add any extra parameters *e.g.* key parameters, during the training time. Because we utilize the TF-IDF machine learning model, it could save the computation load and time without training extra parameters. (2). During the testing stage, we also utilize the TF-IDF machine learning model to extract the instruction token. Compared to L2P and EProj, which need heavy LLMs to embed the input embeddings, our method is small and lightweight. (3). In our codebook, it saves the previous instruction texts (˜KB). While for L2P and EProj, they save the key parameters (˜MB). Thus, we can observe that our method would occupy less memory space.

## H. Term Frequency-Inverse Document Frequency

Term Frequency-Inverse Document Frequency (TF-IDF) (Christian et al., 2016) is a machine learning method commonly used in natural language processing and information retrieval to evaluate the importance of a word in a document relative to a corpus. It combines two components: Term Frequency (TF) and Inverse Document Frequency (IDF). Term Frequency measures how often a term appears in a document:

$$TF(t, d) = \frac{\text{Number of occurrences of term } t \text{ in document } d}{\text{Total number of terms in document } d}. \tag{46}$$

Inverse Document Frequency reduces the weight of terms that in many documents, emphasizing terms in fewer documents:

$$IDF(t, D) = \log \frac{|D|}{1 + |\{d \in D : t \in d\}|}, \tag{47}$$

where $|D|$ is total number of documents in the corpus and $|\{d \in D : t \in d\}|$ is number of documents containing term $t$.

The TF-IDF score is calculated by multiplying TF and IDF:

$$TF\text{-}IDF(t, d, D) = TF(t, d) \cdot IDF(t, D). \tag{48}$$

TF-IDF is effective for capturing term importance and extracting features for text classification, making it a simple yet powerful tool in text processing. In our code, we utilize the TfidfVectorizer class in sklearn library to tokenize the instruction texts into numerical vector.

## I. Instruction Grouping Results

In this section, we summarize the details of three kinds of instruction templates, as shown in Table 9. The instruction grouping results are shown in Table 10. It can be observed that in the three kinds of instruction templates, our strategy divides the instructions of the eight tasks into 4 groups (both for Origin and 10Type), 5 groups (for Diverse), respectively, which can be deemed as a limited expansion compared to the total number of tasks.

## J. Training Details

In the implementation of our method, the codebase is based on CoIN (Chen et al., 2024a). The inserted LoRA in each module layer of LLM has a rank of 128. For each fine-tuning dataset, the training epoch is set to 1, and the initial learning rate and weight decay are configured at 2e-4 and 0. The max length of input text is fitted as 2048. Additionally, we adopt gradient checkpoint strategy and mixed precision mode of TF32 and BF16. Furthermore, we also utilize the ZeRO stage: 0 mode of DeepSpeed for training.

## K. Compared Methods

**LoRA Fine-Tune** (Hu et al., 2022) prepends LoRA parameter efficient tuning paradigm into LLM. In the training stage, it only trains the linear projector and LoRA parameters, with frozen vision encoder and LLM; **MoELoRA** (Chen et al., 2024a) is based on the LoRA, and the number of experts for each MoE layer is set to 2; **LWF** (Li & Hoiem, 2017) calculates the results of the new dataset samples on both the old and new models. After that, it calculates the distillation loss and adds it to the loss function as a regularization penalty term. **EWC** (Kirkpatrick et al., 2017) considers the change of the training parameters and proposes the specific parameters changing loss as a regularization penalty. **PGP** (Qiao et al., 2024a) introduces a gradient projection method for efficient parameters, and changes the gradient direction orthogonal to the previous feature subspace. **E-Proj** (He et al., 2023) is the representative SOTA dynamic modal method, which expands the visual projection layer with the task increasing. **MT** (Zhu et al., 2024) is the representative SOTA regularization method, which compensates for changes in trainable parameters.

## L. Evaluation Metrics

It is worth noting that our judgment of whether the prediction results are correct or not is strictly based on the direct comparison between outputs of MLLMs and ground truths, which is defined as **Truth Alignment** in (Chen et al., 2024a). Therefore, our judgment criteria would be more stringent.

**Average Accuracy** (Avg.ACC) is used for averaging the test accuracy of all datasets, which represents the comprehensive performance of continual tuning.

**Forgetting** (FOR) is utilized to indicate the test accuracy reduction of past datasets after learning the new dataset, which denotes the stability performance.

**New Accuracy** (New.ACC) is employed to average the test accuracy of new datasets, which means the plasticity performance.

Overall, Average Accuracy, Forgetting, and New Accuracy are generally defined as:

$$\text{Average Accuracy} = \frac{1}{T} \sum_{i=1}^{T} A_{T,i}, \tag{49}$$

$$\text{Forgetting} = \frac{1}{T-1} \sum_{i=1}^{T-1} \max(A_{j,i})_{j \in [i, T-1]} - A_{T,i}, \tag{50}$$

$$\text{New Accuracy} = \frac{1}{T} \sum_{i=1}^{T} A_{i,i}, \tag{51}$$

where $T$ is the number of datasets, $A_{T,i}$ is the accuracy of $i$-th dataset on the model trained after $T$-th dataset, $A_{j,i}$ is the accuracy of $i$-th dataset on the model trained after $j$-th dataset, and $A_{i,i}$ is the accuracy of $i$-th dataset on the model trained after $i$-th dataset.

## M. Continual Instruction Tuning Results on Large Language Model

**Dataset.** We use the InstrDialog Stream dataset from (Zhang et al., 2023), which consists of 4 tasks from dialogue state tracking, 11 tasks from dialogue generation, and 4 tasks from intent identification, resulting in a total of 19 dialogue tasks.

**Model.** We use the instruction-tuned T5 model from (Zhang et al., 2023), which has learned to understand some instructions and can act as a good starting point to conduct subsequent learning.

**Comparison Methods.** (1). L2 and EWC regularization methods, both of which leverage a Fisher information matrix to mitigate forgetting by regularizing the loss function, thereby penalizing changes to the crucial parameters of previous tasks. (2). Replay method, which stores random instances from each task in a memory buffer, trains the model jointly on both the new task data and the stored instances. (3). AGEM introduces constraints to ensure that parameter updates do not lead to increased loss on previously learned tasks. These losses are calculated using samples stored in memory. (4). AdapterCL remains the pre-trained model frozen, and trains an independent Adapter for each task. (5). For ours, we freeze the pre-trained model and train a set of Adapters. Specifically, we utilize the instruction grouping strategy to decide whether

*Table 8.* Performance of different methods on the **InstrDialog** dataset. Means and standard deviations are reported as $Mean_{SD}$. † means zero-shot performance.

| Method | InstrDialog | | |
| --- | --- | --- | --- |
| | AR | FWT | BWT |
| FT-no-init | $29.6_{2.1}$ | $8.0_{0.2}$ | $-10.8_{2.3}$ |
| AdapterCL | $8.1_{0.1}$ | $9.4_{0.7}$ | $-21.9_{0.9}$ |
| Init | $22.5^{\dagger}$ | - | - |
| FT-init | $35.7_{0.2}$ | $18.5_{0.7}$ | $-4.6_{0.2}$ |
| L2 | $35.6_{0.1}$ | $17.5_{0.5}$ | $-3.8_{1.2}$ |
| EWC | $34.5_{0.6}$ | $16.8_{0.4}$ | $-6.8_{1.5}$ |
| AGEM (10) | $33.2_{0.4}$ | $19.1_{0.1}$ | $-7.3_{1.0}$ |
| AGEM (50) | $34.9_{0.9}$ | $18.1_{1.0}$ | $-6.0_{0.9}$ |
| Replay (10) | $38.4_{0.7}$ | $23.7_{0.0}$ | $-1.3_{0.5}$ |
| Replay (50) | $40.4_{0.0}$ | $22.9_{0.1}$ | $1.6_{1.2}$ |
| **Ours** | $\mathbf{39.8}_{0.5}$ | $\mathbf{21.3}_{0.2}$ | $\mathbf{0.8}_{0.3}$ |
| Multi | $42.1_{0.6}$ | - | - |

to retrain or expand the Adapter according to the current task instructions and adopt dynamical EMA update when retraining the Adapter. Notice here, besides AdapterCL and ours, which freeze the LLM and only train the Adapter, others adopt the full fine-tuning (FFT) mode.

**Metrics.** Following (Zhang et al., 2023), we use the following metrics to measure the continual tuning performance. Let $a_{j,i}$ represents the ROUGE-L score obtained by the model on the test set of task $t_i$ after it has been trained on task $t_j$. We define the following metrics:

**Average *ROUGE-L*** (AR), evaluates the model's average performance across all tasks after completing the training on the task $t_T$:

$$\mathbf{AR} = \frac{1}{T} \sum_{i=1}^{T} a_{T,i}. \tag{52}$$

**Forward Transfer** (FWT), evaluates the extent to which knowledge from previous tasks facilitates learning a new task:

$$\mathbf{FWT} = \frac{1}{T-1} \sum_{i=2}^{T} a_{i-1,i}. \tag{53}$$

**Backward Transfer** (BWT), evaluates the impact that continually learning on subsequent tasks has on previous tasks:

$$\mathbf{BWT} = \frac{1}{T-1} \sum_{i=1}^{T-1} \left( a_{T,i} - a_{i,i} \right). \tag{54}$$

It is worth noting that a positive BWT value indicates that the performance of previous tasks can be improved by subsequent tasks, whereas a negative value signifies knowledge forgetting.

From Table 8, we can observe that (1). Our method surpasses all other non-replay methods, as well as the replay method with a memory size of 10, and the AGEM method in terms of the $AR$ metric. While it is only slightly behind the replay method with a memory size of 50 (39.8 *vs.* 40.4). These results demonstrate its strong continual instruction tuning performance, with achieving results comparable to those of replay methods (note that our method is a non-replay method).

(2) For the $BWT$ metric, all other non-replay methods own negative values. However, because of the EMA-based ability to review the old knowledge, our method aligns with the replay method with a memory size of 50 (positive $BWT$ values), indicating excellent anti-forgetting capability.

(3) In terms of the $FWT$ metric, since our method achieves an optimal balance between stability and plasticity, its performance is highly competitive compared to other methods.

## N. Three Types of Tuning Order Sequences

In order to validate the robustness of our method, we adopt the following three types of tuning order.

1). Origin tuning order: ScienceQA, TextVQA, ImageNet, GQA, VizWiz, Grounding, VQAv2, OCR-VQA.

2). Reverse tuning order: OCR-VQA, VQAv2, Grounding, VizWiz, GQA, ImageNet, TextVQA, ScienceQA.

3). Alphabet tuning order: GQA, Grounding, ImageNet, OCR-VQA, ScienceQA, TextVQA, VizWiz, VQAv2.

## O. Three Types of Instruction Templates

In order to validate the robustness of our method, we adopt the following three types of training instructions. For detailed instruction templates please kindly refer to Table 9.

1). Original instruction type: Each task owns only one instruction, and several tasks share the same instructions.

2). Diverse instruction type: Each task owns only one instruction, and different tasks are tailored to distinct instructions.

3). 10Type instruction type: Each task owns around ten instructions, and several tasks share similar instructions.

## P. Deduction of high-dimensional $\Theta_t$

For high-dimensional $\Theta_t$, we have the following Taylor expansion results:

$$\mathcal{L}(\Theta) = \mathcal{L}(\Theta_t) + \left(\frac{\partial \mathcal{L}}{\partial \Theta}\right)^T (\Theta - \Theta_t) + (\Theta - \Theta_t)^T \frac{H}{2}(\Theta - \Theta_t) + O(\Theta - \Theta_t)^3. \tag{55}$$

To further introduce $\Theta_t^*$ in the Taylor expansion, we replace $\Theta$ with $\Theta_t^*$, and have:

$$\mathcal{L}(\Theta_t^*) - \mathcal{L}(\Theta_t) = \left(\frac{\partial \mathcal{L}}{\partial \Theta}\right)^T \bigg|_{\Theta=\Theta_t^*} (\Theta_t^* - \Theta_t) + (\Theta_t^* - \Theta_t)^T \frac{H}{2}(\Theta_t^* - \Theta_t). \tag{56}$$

Notice that we have omitted the high-order infinitesimal term of $O(\Theta - \Theta_t)^3$. In this way, we can obtain the following minimum optimization goal:

$$F = \left(\frac{\partial \mathcal{L}}{\partial \Theta}\right)^T \bigg|_{\Theta=\Theta_t^*} (\Theta_t^* - \Theta_t) + (\Theta_t^* - \Theta_t)^T \frac{H}{2}(\Theta_t^* - \Theta_t) + e^T(\Theta_t^* - \Theta_{t-1}^*) + \lambda e^T(\Delta\Theta + \Theta_{t-1}^* - \Theta_t^*). \tag{57}$$

The following deduction is similar to the process of $\theta$, finally, we have:

$$0 = \frac{1}{(\beta_t - 1)} \left(\frac{\partial \mathcal{L}}{\partial \Theta}\right)^T \bigg|_{\Theta=\Theta_t^*} \Delta\Theta + \frac{\beta_t}{(\beta_t - 1)^2} \Delta\Theta^T H \Delta\Theta + \frac{e^T \Delta\Theta}{\beta_t - 1}. \tag{58}$$

Considering that $\Delta\Theta = (\Theta_{t-1}^* - \Theta_t)(\beta_t - 1)$, we can obtain:

$$0 = \left(\frac{\partial \mathcal{L}}{\partial \Theta}\right)^T \bigg|_{\Theta=\Theta_t^*} (\Theta_{t-1}^* - \Theta_t) + \beta_t(\Theta_{t-1}^* - \Theta_t)^T H(\Theta_{t-1}^* - \Theta_t) + e^T(\Theta_{t-1}^* - \Theta_t). \tag{59}$$

Obviously, $\beta_t$ equals to:

$$\beta_t = [\left(\frac{\partial \mathcal{L}}{\partial \Theta}\right)^T \bigg|_{\Theta=\Theta_t^*} + e^T](\Theta_t - \Theta_{t-1}^*) \cdot [(\Theta_t - \Theta_{t-1}^*)^T H(\Theta_t - \Theta_{t-1}^*)]^{-1}. \tag{60}$$

It can be discovered that Eq.(60) has the same form as Eq.(12). Thus, we can draw the conclusion that our method can be expanded to high-dimensional $\Theta_t$.

*Table 9.* The list of instructions for each task.

| Task | Original | Diverse | 10Type |
|---|---|---|---|
| **ScienceQA** | Answer with the option's letter from the given choices directly | Answer with the option's letter from the given choices directly | Answer with the option's letter from the given choices directly
Select the correct answer from the given choices and respond with the letter of the chosen option
Determine the correct option from the provided choices and reply with its corresponding letter
Pick the correct answer from the listed options and provide the letter of the selected option
Identify the correct choice from the options below and respond with the letter of the correct option
From the given choices, choose the correct answer and respond with the letter of that choice
Choose the right answer from the options and respond with its letter
Select the correct answer from the provided options and reply with the letter associated with it
From the given choices, select the correct answer and reply with the letter of the chosen option
Identify the correct option from the choices provided and respond with the letter of the correct option
From the given choices, pick the correct answer and respond by indicating the letter of the correct option |
| **TextVQA** | Answer the question using a single word or phrase | Capture the essence of your response in a single word or a concise phrase | Answer the question with just one word or a brief phrase
Use one word or a concise phrase to respond to the question
Answer using only one word or a short, descriptive phrase
Provide your answer in the form of a single word or a brief phrase
Use a single word or a short phrase to respond to the question
Summarize your response in one word or a concise phrase
Respond to the question using a single word or a brief phrase
Provide your answer in one word or a short, descriptive phrase
Answer the question with a single word or a brief, descriptive phrase
Capture the essence of your response in one word or a short phrase
Capture the essence of your response in a single word or a concise phrase |
| **ImageNet** | Answer the object in the image using a single word or phrase | Express the object in the image in a single word or a short, descriptive phrase | Summarize the object in the image in a single word or a brief phrase
Provide the object in the image using a single word or a brief phrase
Give the object in the image in the form of a single word or a concise phrase
Express the object in the image with one word or a short, descriptive phrase
Identify the type of content in the image using one word or a concise phrase
Respond to the object in the image with a single word or a short, descriptive phrase
Describe the content of the image using one word or a concise phrase
Express the object in the image in a single word or a short, descriptive phrase
Use a single word or a short phrase to categorize the image content
Classify the image content using only one word or a brief phrase
Use one word or a short phrase to classify the content of the image |
| **GQA** | Answer the question using a single word or phrase | Respond to the question briefly, using only one word or a phrase | Respond to the question with a single word or a short phrase
Respond to the question using only one word or a concise phrase
Answer the question with a single word or a brief phrase
Respond with one word or a short phrase
Provide your answer in the form of a single word or a concise phrase
Respond to the question with just one word or a brief phrase
Answer the question using a single word or a concise phrase
Provide your response using only one word or a short phrase
Respond to the question with a single word or a brief phrase
Respond to the question using just one word or a concise phrase
Answer the question with one word or a short phrase |
| **VizWiz** | Answer the question using a single word or phrase | Provide a succinct response with a single word or phrase | Answer the question using only one word or a concise phrase
Respond to the question using only one word or a concise phrase
Respond to the question with a single word or a brief phrase
Provide your answer using just one word or a short phrase
Respond with one word or a concise phrase
Answer the question with just one word or a brief phrase
Use a single word or a short phrase to answer the question
Provide your answer in the form of one word or a brief phrase
Reply to the question using one word or a concise phrase
Answer with a single word or a short phrase
Use one word or a brief phrase to answer the question |
| **Grounding** | Please provide the bounding box coordinate of the region this sentence describes | Please provide the bounding box coordinate of the region this sentence describes | Identify and provide the bounding box coordinates that match the description given in this sentence
Extract and provide the bounding box coordinates based on the region described in the sentence
Please provide the bounding box coordinate of the region this sentence describes
Find and provide the bounding box coordinates for the region mentioned in the sentence
Provide the coordinates of the bounding box that correspond to the region described in the sentence
Give the bounding box coordinates as described in the sentence
Determine and provide the bounding box coordinates based on the description in the sentence
Identify and provide the coordinates of the bounding box described in the sentence
Provide the coordinates for the bounding box based on the region described in the sentence
Extract and provide the coordinates for the bounding box described in the sentence
Identify and give the coordinates of the bounding box as described by the sentence |
| **VQAv2** | Answer the question using a single word or phrase | Answer the question using a single word or phrase | Answer the question using a single word or phrase
Answer the question with a single word or a brief phrase
Use one word or a short phrase to respond to the question
Answer the question using just one word or a concise phrase
Provide your answer to the question using only one word or a brief phrase
Respond to the question with a single word or a short phrase Use a single word or phrase to answer the question
Provide an answer using only one word or a brief phrase
Answer the question succinctly with one word or a brief phrase
Answer the question with just one word or a short phrase
Respond to the question using a single word or a concise phrase |
| **OCR-VQA** | Answer the question using a single word or phrase | Condense your answer for each question into a single word or concise phrase | Respond to the question with a single word or a short phrase
Answer the question using a single word or a concise phrase
Provide your response using only one word or a short phrase
Use one word or a brief phrase to answer the question
Reply to the question using one word or a concise phrase
Use a single word or a short phrase to answer the question
Use a single word or phrase to answer the question
Provide an answer using only one word or a brief phrase
Provide your answer to the question using only one word or a brief phrase
Respond to the question using a single word or a concise phrase
Answer the question using a single word or phrase |

Table 10. The instruction grouping results.

| Type | Group 1 | Group 2 | Group 3 | Group 4 | Group 5 |
|---|---|---|---|---|---|
| **Origin** | Answer with the option's letter from the given choices directly | Answer the question using a single word or phrase | Answer the object in the image using a single word or phrase | Please provide the bounding box coordinate of the region this sentence describes | - |
| **Diverse** | Answer with the option's letter from the given choices directly | Capture the essence of your response in a single word or a concise phrase / Provide a succinct response with a single word or phrase | Express the object in the image in a single word or a short, descriptive phrase | Respond to the question briefly, using only one word or a phrase / Answer the question using a single word or phrase / Condense your answer for each question into a single word or concise phrase | Please provide the bounding box coordinate of the region this sentence describes |
| **IOType** | Answer with the option's letter from the given choices directly
Select the correct answer from the given choices and respond with the letter of the chosen option
Determine the correct option from the provided choices and reply with its corresponding letter
Pick the correct answer from the listed options and provide the letter of the selected option
Identify the correct choice from the options below and respond with the letter of the correct option
From the given choices, choose the correct answer and respond with the letter of that choice
Choose the right answer from the options and respond with its letter
Select the correct answer from the provided options and reply with the letter associated with it
From the given choices, select the correct answer and reply with the letter of the chosen option
Identify the correct option from the choices provided and respond with the letter of the correct option
From the given choices, pick the correct answer and respond by indicating the letter of the correct option | Use a single word or short phrase to respond to the question
Provide your answer in one word or a short, descriptive phrase
Summarize your response in one word or a concise phrase
Answer using only one word or a short, descriptive phrase
Answer the question with a single word or a brief, descriptive phrase
Use one word or a concise phrase to respond to the question
Capture the essence of your response in a single word or a concise phrase
Capture the essence of your response in one word or a short phrase
Answer the question with just one word or a brief phrase
Provide your answer in the form of a single word or a brief phrase
Respond to the question using a single word or a brief phrase
Provide your answer in the form of a single word or a concise phrase
Respond to the question with a single word or a short phrase
Answer the question using a single word or a concise phrase
Respond to the question with a single word or a brief phrase
Provide your response using only one word or a concise phrase
Answer the question with one word or a short phrase
Respond with one word or a short phrase
Answer the question with a single word or a brief phrase
Respond to the question using just one word or a concise phrase
Provide your answer in the form of one word or a brief phrase
Provide your answer using just one word or a short phrase
Use one word or a brief phrase to answer the question
Reply to the question using one word or concise phrase
Use a single word or one word or concise phrase
Answer the question using a single word or phrase
Use one word or a short phrase to respond to the question
Answer the question succinctly with one word or a brief phrase
Use a single word or phrase to answer the question
Provide an answer using only one word or a brief phrase
Provide your answer to the question using only one word or a brief phrase
Answer the question with just one word or a concise phrase
Answer the question using just one word or a concise phrase
Respond to the question using a single word or a concise phrase
Answer the question using a single word or phrase | Summarize the object in the image in a single word or a brief phrase
Provide the object in the image using a single word or a brief phrase
Give the object in the image in the form of a single word or a concise phrase
Express the object in the image in one word or a short, descriptive phrase
Identify the type of content in the image using one word or a concise phrase
Respond to the object in the image with a single word or a short, descriptive phrase
Describe the content of the image using one word or a concise phrase
Express the object in the image in a single word or a short, descriptive phrase
Use a single word or a short phrase to categorize the image content
Classify the image content using only one word or a brief phrase
Use one word or a short phrase to classify the content of the image | Extract and provide the coordinates for the bounding box described in the sentence
Please provide the bounding box coordinate of the region in this sentence describes
Give the bounding box coordinates as described in the sentence
Extract and provide the bounding box coordinates based on the region described in the sentence
Find and provide the bounding box coordinates for the region mentioned in the sentence
Identify and provide the bounding box coordinates that match the description given in this sentence
Determine and provide the bounding box coordinates based on the description in the sentence
Provide the coordinates for the bounding box based on the region described in the sentence
Provide the coordinates of the bounding box that correspond to the region described in the sentence
Identify and provide the coordinates of the bounding box described in the sentence
Identify and give the coordinates of the bounding box as described by the sentence | - |

# Q. Algorithm

---

**Algorithm 1** Dynamical EMA Updating and Instruction Grouping

---

**Input:** Pre-trained LFMs $f_{lfm}$, number of datasets $D$, number of iterations $T$, training set $\{\{x_i^t, I_i^t, y_i^t\}_{i=1}^{n_t}\}_{t=1}^{T}$, learning rate $\eta$, loss function $\mathcal{L}_x$, matching threshold $\epsilon$.
**Output:** training parameters pool $\{f_{trn}^i\}_{i=1}^{n}$, instruction codebook $\{I^i\}_{i=1}^{n}$.
**initialize:** $\{f_{trn}^i\}_{i=1}^{n}$, $\{I^i\}_{i=1}^{n}$.
 **for** $d = 1, ..., D$ **do**
   1. Collect the instructions of the current task $I_c$, match with $I^i$ in codebook, and obtain the maximum cosine similarity $s$.
    **if** $s \geq \epsilon$ **then**
    | 2. Initialize the current training parameters $f_{trn}$ and EMA parameters $f_{trn}^*$ from the matching parameters.
    **else**
    | 2. Initialize the current training parameters $f_{trn}$ from the last EMA parameters $f_{trn}^*$.
    **end**
   **for** $epoch = 1$ **do**
     **for** $t = 1, ..., T$ **do**
       3. Draw a mini-batch $B = \{(x_i^t, I_i^t, y_i^t)\}_{i=1}^{n_t}$.
      **for** $(x_t, I_t, y_t)$ *in* $B$ **do**
        **if** $s \geq \epsilon$ **then**
          4. Prepend the $f_{trn}$ into the $f_{lfm}$ and obtain prediction $\hat{y}_t = f_{lfm}([x_t; I_t])$.
          5. Calculate per batch loss $\mathcal{L}_B$ by accumulating $\mathcal{L}_x(y, \hat{y})$ and update $f_{trn}$ with optimizer.
          6. Record $f_{trn}$ and the corresponding gradients $\mathcal{L}'$ at iteration $t$.
          7. Calculate EMA weight $\beta_t$ according to Eq.(45).
          8. Update $f_{trn}^*$ by Eq.(1).
          9. Clear $f_{trn}$ and $\mathcal{L}'$ at iteration $t-1$.

        **else**
          4. Prepend the $f_{trn}$ into the $f_{lfm}$ and obtain prediction $\hat{y}_t = f_{lfm}([x_t; I_t])$.
          5. Calculate per batch loss $\mathcal{L}_B$ by accumulating $\mathcal{L}_x(y, \hat{y})$ and update $f_{trn}$ with optimizer.

        **end**
      **end**
     **end**
   **end**
   **if** $s \geq \epsilon$ **then**
   | 10. Cover matching parameters in the pool with $f_{trn}^*$ and update the codebook with $I_c$.
   **else**
   | 6. Append $f_{trn}$ into the pool and update the codebook with $I_c$.
   **end**
**end**

