# OpenReview forum: "Large Continual Instruction Assistant"
_ICML.cc/2025/Conference — ICML 2025 poster_

### Official Review · Reviewer_4GQq · 2025-03-03

**Overall Recommendation:** 4

**Summary:**

To address catastrophic forgetting in Large Foundation Models (LFMs), this paper proposes a general continual instruction tuning framework. It introduces a dynamic exponential moving average update method to preserve prior knowledge while assimilating new information. For realizing the balance of stability and plasticity in LFMs, this paper derives a self-adaptive EMA weight for each update process. Furthermore, an instruction grouping strategy allows for retraining parameters of semantically similar instructions and selectively expanding those of semantically divergent ones. Experiments on distinct MLLMs and LLMs, including multimodal continual instruction benchmark and language continual instruction benchmark, all demonstrate the framework's strong resistance to forgetting and excellent continual instruction tuning performance.

**Claims And Evidence:**

All the claims made in the submission are supported by clear and convincing evidence.

**Essential References Not Discussed:**

This paper has already included all the essential references.

**Experimental Designs Or Analyses:**

Although the experimental designs in this paper are comprehensive and the analyses are sound, the authors may further enhance the completeness of the experiment from the following perspectives:
1. In Table 3, considering to include additional state-of-the-art methods would allow for a more comprehensive evaluation of the proposed method on the QWen-VL architecture.
2. Compared to the method with stable EMA weight, the additional dynamic adjustment time required by the proposed self-adaptive EMA weight remains unknown. It is recommended that the authors compare training time on a specific dataset with stable/dynamic EMA update methods to help readers better assess the efficiency of the proposed method.

**Methods And Evaluation Criteria:**

The proposed methods and used evaluation criteria make sense for the problem or application at hand.

**Other Comments Or Suggestions:**

The use of some terms in the paper is not consistent enough, such as "forgetting" sometimes referring to "catastrophic forging" and sometimes referring to "forging metric". The authors are suggested to distinguish between these terms for clarity.

**Other Strengths And Weaknesses:**

Strengths
+ The experimental results in this paper demonstrate significant improvements in both anti-forgetting and continual tuning performance, especially on the LLaVA-7B and LLaVA-13B models, which outperform existing state-of-the-art methods.
+ The instruction grouping strategy utilizes the TF-IDF model for instruction matching, avoiding complex embedding calculations, reducing the computational burden and computational costs.
+ The proposed method exhibits good robustness and generality, making it applicable to various continual instruction tuning scenarios and large foundation models.

Weaknesses
- This paper contains several redundant phrases. Such as in line 412-414, "Additionally, we are surprised to find that our method can spontaneously suppress the occurrence of hallucinations appearing in the continual instruction tuning,", the phrase 'appearing in' is redundant and should be revised to: "Additionally, we are surprised to find that our method can spontaneously suppress the occurrence of hallucinations in the continual instruction tuning.".
- This paper contains terminology issues. Such as in line 90-92, "Our method is model-free and can be easily applied to a wide range of CIT methods." should be modified as "Our method is model-agnostic and can be easily applied to a wide range of CIT methods.", which would be more accurate.

**Questions For Authors:**

This paper only provides the instruction reuse results on MLLM architectures (Table 10). While the instruction reuse results on LLM are unknown.

**Relation To Broader Scientific Literature:**

The paper effectively cites research in related fields, particularly the latest developments in continual instruction tuning for Large Foundation Models. In contrast to the related scientific literature, the primary contributions of this paper are as follows:
1. Compared to the other former methods, this paper examines the catastrophic forgetting issue in continual instruction tuning from the perspective of the plasticity-stability trade-off in Large Foundation Models, which is the core of continual learning. Thorough rigorous mathematical derivation, the paper presents compelling results.
2. By exploring and exploiting the phenomenon of instruction reuse, this paper employs lightweight machine learning algorithm (TF-IDF) to cluster semantically similar instructions, enabling limited model expansion. In comparison to other model expansion techniques like L2P [1] and EProj [2], this approach significantly improves training efficiency while reducing computational and memory overhead.

[1] Learning to prompt for continual learning
[2] Continual instruction tuning for large multimodal models

**Theoretical Claims:**

I checked the correctness of proofs as followings:
1. Equation (3) to Equation (12) and Equation (28) to Equation (39) are based on Taylor expansion and Lagrange multiplier method, deriving the optimal solution for EMA weights.
2. Equation (13) to Equation (15) and Equation (43) to Equation (45) employ several well-founded mathematical approximations, which obtain the highly accurate approximate solution, with effectively reducing both complexity and computational cost.

---

> ### Author Rebuttal · Authors · 2025-03-31
>
> Dear reviewer 4GQq, thanks for your valuable suggestions. Here are our responses:
>
> **Response 1 (QWen Baselines)**: We newly implemented another two strong baselines: EProj and EWC on the QWen-VL architecture. The results are presented as:
>
> |Method|Venue|ScienceQA|TextVQA|ImageNet|GQA|VizWiz|Grounding|VQAv2|OCRVQA|Avg.ACC|Forgetting|New.ACC|
> |:------:|:------:|:-------:|:-------:|:-------:|:-------:|:-------:|:-------:|:-------:|:-------:|:-------:|:--------:|:-------:|
> |LoRA|ICLR’22|31.05|42.45|29.57|55.57|15.30|40.33|67.75|47.80|41.23|19.36|58.17|
> |EWC|PNAS’17|64.30|58.67|44.04|57.73|38.16|48.04|66.98|41.76|52.46|8.68|50.67|
> |PGP|ICLR’24|66.42|41.33|32.16|49.83|36.05|24.22|58.60|43.96|44.07|5.90|48.30|
> |EProj|ArXiv’24|63.19|59.28|62.96|55.51|40.69|41.20|66.89|45.30|54.38|2.52|56.59|
> |**Ours**|**-**|**66.52**|**59.44**|**53.56**|**57.81**|**39.57**|**47.44**|**70.36**|**50.44**|**55.64**|**1.62**|**56.19**|
>
> The results demonstrate that our method still owns the best performance, surpassing other state-of-the-art approaches.
>
> **Response 2 (Resource Consumption)**: Please kindly refer to the **Response 1** for the same question of Reviewer pe71.
>
> **Response 3 (Phrase)**: In the revised version, we have modified the sentence in lines 412-414 as follows: "Additionally, we are surprised to find that our method can spontaneously suppress the occurrence of hallucinations in continual instruction tuning." We also conducted a thorough revision of the paper to identify and eliminate other redundant expressions, ensuring the writing remains clear and precise.
>
> **Response 4 (Terminology)**: We have revised lines 90-92 as follows: "Our method is model-agnostic and can be easily applied to a wide range of CIT methods." Additionally, we carefully reviewed the manuscript to ensure consistent and accurate use of technical terminology throughout the paper.
>
> **Response 5 (Expression)**: To address this, we have clarified the distinction between the "forgetting metric" and "catastrophic forgetting" by adding the explanation that, in our paper, "Forgetting" (with an uppercase "F") refers to the "forgetting metric," whereas "forgetting" (with a lowercase "f") denotes the "catastrophic forgetting.”
>
> **Response 6 (Instruction Grouping Results on LLM)**: To address your concern, we have presented the instruction grouping results on LLM as follows.
>
> **Group 1**
>
> pubmedqa_classification = ... Output '1' if the passage has a defininte objective/aim/goal and output '0' if the passage does not have a definite objective/aim/goal …
>
> dstc3_classification = ... output the price range the user if looking for which can take one of four values: Cheap, Moderate, Expensive and Don't Care …
>
> air_dialogue_classification = ... select the goal of the conversation …
>
> deal_or_no_dialog_classification = ... answer 'Yes' if both participants agree to the deal, otherwise answer 'No' …
>
> craigslist_bargains_classification = ... classify the text into one of the labels from the two possible outputs - 'accepted'/'rejected' …
>
> **Group 2**
>
> mutual_multi_turn_dialogue = ... choose the most reasonable option …
>
> personachat_choose_next = ... Choose one and answer with the text …
>
> **Group 3**
>
> circa_answer_generation = ... generate an answer that is relevant to the question …
>
> convai3_sentence_generation = ... read the input, then generate a valid prediction of the user's response to the computer's clarifying question …
>
> air_dialogue_sentence_generation = ... find the answer of the previous dialogue …
>
> diplomacy_text_generation = ... generate the next message …
>
> curiosity_dialogs_answer_generation = ... find the dialogue that is basically a response given to a question or an aspect of the user ...
>
> smcalflow_sentence_generation = ... identify what will be users' command for that reply ...
>
> multi_woz_user_utterance_generation = ... Generate a language query such that it leads to this reply ...
>
> personachat_generate_next = ... generate the next utterance in a given dialogue ...
>
> dstc3_answer_generation = ... answer the given question based on the information present in the dialogue ...
>
> convai3_sentence_generation = ... generate a prediction of what the requester is actually trying to do ...
>
> smcalflow_sentence_generation = ... return what will be Agent's response/reply for that particular user's command or question ...
>
> **Group 4**
>
> storycommonsense_motiv_text_generation = ... write the character's motivation by doing a specific job, which is given in the sentence …
>
> It can be seen that **instructions for 19 tasks are categorized into 4 groups**, which reflects the limited model expansion property of our method.
>
> Thanks for your detailed and constructive review. Your review has significantly contributed to the refinement of our work, offering new insights and helping us address important aspects that were previously overlooked. We have made extensive revisions based on your suggestions, resulting in a clearer and more robust paper.

---

### Official Review · Reviewer_9qib · 2025-03-03

**Overall Recommendation:** 4

**Summary:**

This paper proposes a novel framework to address the catastrophic forgetting in Continual Instruction Tuning (CIT). Starting from the trade-off between the plasticity and stability, the paper introduces an optimal balance weight of Exponential Moving Average (EMA), determined automatically by gradients and learned parameters. Additionally, the framework uses semantic similarity of instructions to decide whether to retrain or expand the model’s training parameters, and allocate the most suitable parameters to testing instances. Extensive experiments across multiple CIT benchmarks (e.g. CoIN, InstrDialog) and Large Foundation Models (LFMs, e.g. LLaVA, QWen-VL, T5) demonstrate that the proposed method not only reduces forgetting but also significantly improves overall continual tuning performance.


## update after rebuttal

The authors have adequately addressed my concerns. My overall recommendation remains accept.

**Claims And Evidence:**

Yes, all the claims are supported by clear and convincing evidence.

**Essential References Not Discussed:**

The paper has discussed almost all essential references.

**Experimental Designs Or Analyses:**

I have checked the soundness and validity of the experimental designs and analyses. The following two issues can be considered to further improve the paper’s quality.
1. In Table 6, the paper presents an ablation study results. However, it only compares results using a fixed weight of 0.99 to show the superiority of proposed Dynamical EMA method. I suggest adding additional results with other fixed weights and providing more evaluations.
2. In Table 3, compared baselines conducted on the QWen-VL architecture are insufficient, adding more state-of-the-art method with stronger baselines like PGP could be employed for further studies of the method.

**Methods And Evaluation Criteria:**

Yes, the proposed methods and evaluation criteria make sense.

**Other Comments Or Suggestions:**

1. In line 96-97, it has “especially significant enhancements in the performance of the baseline (as shown in Figure 1)”. Here, the author may clarify the “baseline” to a more detailed “LoRA Fine-tuning baseline”.

**Other Strengths And Weaknesses:**

**Strengths**

1.	This paper presents extensive experiments across multiple Large Foundation Models, and several benchmark datasets, as well as natural language processing tasks. The experimental results demonstrate that the proposed method outperforms existing state-of-the-art methods in terms of anti-forgetting and continual tuning performance.
2.	This paper proposes a method for dynamically adjusting EMA weights, theoretically deriving the optimal EMA weights to balance stability and plasticity in the continual learning process. Compared to traditional fixed EMA weight methods, the dynamic adjustment mechanism better adapts to continuously changing datasets and significantly mitigates catastrophic forgetting.
3.	This paper proposes a semantic similarity-based instruction grouping strategy that clusters similar instructions using the TF-IDF model and assigns a trainable LoRA set to each instruction group. Compared to existing model expansion methods, this strategy not only limits the size of expansion parameters but also reduces computational costs.

**Weaknesses**

1. Some tables’ title (e.g., Table 6) are too brief; it is recommended to add more detailed explanatory sentences for clarity.
2. The paper introduces the L1-Norm to approximate $\beta_t$. However, in line 248-251, the authors only state that the motivation for adopting L1-Norm to approximate the $\beta_t$ is L1-Norm occupies a few computation loads compared to other normalization methods. Providing experimental comparisons with other representative Normalization method (e.g. L2-Norm) would be more more convincing.

**Questions For Authors:**

1. In Table 5, it can be observed that the Avg.ACC results for Origin and Diverse instruction types are slightly higher than those for the 10Type instruction type. What might explain this discrepancy? Is it related to the type of instruction template?
2. In Table 1-3, the authors utilize the Forgetting as the evaluation metric. However, in Table 8, they switch to the BWT metric. Why do the authors choose different evaluation metrics to assess the anti-forgetting ability of MLLMs and LLMs?

**Relation To Broader Scientific Literature:**

Compared to broader scientific literature on large model continual instruction tuning (e.g., EProj [1], CoIN [2]), the main contribution of this paper lies in 1. For the first time, this paper addresses the forgetting problem in continual instruction tuning from the perspective of the trade-off between plasticity and stability in Large Foundation Models, with providing strong and convincing theoretical proof, which offers insightful ideas. 2. This paper proposes a dynamic exponential moving average update strategy to balance the trade-off between plasticity and stability, achieving significant and surprising performance improvements. Additionally, the method can be applicable to various Large Foundation Model architectures, including LLMs and MLLMs (e.g., LLaVA, QWen-VL), demonstrating strong generalization. 3. Based on the phenomenon of instruction reuse and lightweight TF-IDF model, the paper achieves limited model expansion. Compared to other model expansion methods like L2P [3] and EProj [1], it effectively improves training efficiency and reduces memory usage, providing further insights into instruction for continual instruction tuning.

[1] Continual instruction tuning for large multimodal models
[2] Coin: A benchmark of continual instruction tuning for multimodel large language model
[3] Learning to prompt for continual learning

**Theoretical Claims:**

I have checked the correctness of all proofs for theoretical claims, including the optimal EMA weight deduction (Eq.(2)-Eq.(15) in the paper and Eq.(28)-Eq.(39) in the supplementary material) and the EMA expansion process (Eq.(16)-Eq.(21) in the supplementary material), etc.

---

> ### Author Rebuttal · Authors · 2025-03-31
>
> Dear reviewer 9qib, thanks for your valuable suggestions. Here are our responses:
>
> **Response 1 (Fixed EMA)**: To address your concern, we have conducted ablation studies with more fixed EMA weights, *i.e.* 0.993, 0.996 0.999 (as the EWA weight is usually set in [0.990, 0.999]) [1]. The results are shown as:
>
> |EMA Weight|Avg.ACC|Forgetting|New.ACC|
> |:--------------------------------:|:-----:|:--------:|:-----:|
> |0.990|48.09|16.24|62.30|
> |0.993|48.78|17.89|64.44|
> |0.996|51.14|15.56|64.76|
> |0.999|51.81|0.85|49.44|
> |Dynamical EMA|55.33|7.04|61.49|
> |Dynamical EMA+Instruction Grouping|64.64|1.93|66.33|
>
> From the Table, our proposed dynamic EMA method consistently achieves the best performance across all three metrics compared to certain fixed EMA weights (e.g., from 0.990 to 0.996). When compared to a fixed EWA weight of 0.999, our method demonstrates superior Avg.ACC and New.ACC but exhibits inferior resistance to forgetting. It is important to clarify that, as described by Equation (1) in our manuscript, a higher EMA weight generally results in lower Forgetting but also lower New.ACC. However, the lower Forgetting does not always indicate better overall performance due to the trade-off between plasticity and stability. In extreme cases, the Avg.ACC metric may be significantly impacted by a reduced New.ACC. As shown in the Table, under the 0.999 EMA weight setting, the New.ACC metric only reaches 49.44, which is considerably lower than the New.ACC values of other methods (others all exceeding 60). In summary, the low Forgetting in the 0.999 EMA weight setting comes at the cost of significantly reduced New.ACC.
>
> **Response 2 (QWen Baselines)**: Please kindly refer to the **Response 1** for the same question of Reviewer 4GQq.
>
> **Response 3 (Table Titles)**: We recognize that some table titles, including Table 6, may be too brief. To improve clarity, we have revised the titles of these tables to include more detailed explanations. For example, we have updated the title of Table 6 to “Ablation study results for each proposed components”.
>
> **Response 4 (L1-Norm)**: We have replaced L1-Norm in our method with L2-Norm and reconducted the whole experiment based on the LLaVA-7B architecture with the Origin instruction type. The results are shown as:
>
> |Normalization|Avg.ACC|Forgetting|New.ACC|
> |:-----------:|:-----:|:--------:|:-----:|
> |L1|64.64|1.93|66.33|
> |L2|61.17|4.08|64.73|
>
> We can see that the L1-Norm consistently outperforms L2-Norm, regardless of Avg.ACC, Forgetting, or New.ACC metrics, which further enhances our motivations of adopting L1-Norm approximation.
>
> **Response 5 (Phrase)**: We have replaced "baseline" with "LoRA Fine-tuning baseline" to specify which baseline is being referred to, making the context clearer for readers. The updated sentence likes: "especially significant enhancements in the performance of the LoRA Fine-tuning baseline (as shown in Figure 1)."
>
> **Response 6 (Avg.ACC Results For Instruction Template)**: As you have pointed out, this phenomenon is closely related to the type of instruction template. As we presented in the Table 9, the 10Type instruction type owns 10 different instruction templates for each task, which require the MLLM to memorize a larger amount of information. Consequently, the challenge of mitigating forgetting is harder compared to the Origin and Diverse instruction types, which utilize only a single instruction template for each task. This conclusion is further supported by the Forgetting values reported in Table 5 (Origin: 1.93, Diverse: 0.45, 10Type: 2.86). Therefore, due to the increased forgetting, the Avg.ACC of the 10Type instruction type is slightly lower than those observed of the Origin and Diverse instruction types.
>
> **Response 7 (Forgetting Metric)**: In fact, both the Forgetting and BWT metrics can be used to measure catastrophic forgetting. The key distinction lies in their typical applications: the BWT metric is more commonly employed in Task Incremental Learning (TIL), whereas the Forgetting metric is often used in Class Incremental Learning (CIL) [1,2]. In our paper, the choice of evaluation metric is guided by the need to ensure consistency with the benchmarks utilized [3,4]. Therefore, we follow their original metric setting in our evaluation.
>
> [1] Trgp: Trust region gradient projection for continual learning.
>
> [2] Learning to prompt for continual learning.
>
> [3] Coin: A benchmark of continual instruction tuning for multimodel large language model.
>
> [4] Citb: A benchmark for continual instruction tuning.
>
> We would like to express our sincere thanks for your thorough review and valuable suggestions. Your thoughtful comments have been crucial in identifying areas for improvement, allowing us to refine and enhance the quality of our paper. Thank you again for your effort and dedication in helping us improve our work.

---

### Official Review · Reviewer_pe71 · 2025-03-14

**Overall Recommendation:** 5

**Summary:**

This paper addresses an important catastrophic forgetting challenge and proposes a solution in the continual instruction tuning. Based on the ideal conditions of balancing plasticity and stability, meanwhile combined with the Exponential Moving Average (EMA) update, the authors adopt the optimization method to obtain the dynamic EMA weight through gradients (new knowledge) and parameters (old knowledge). The dynamic EMA weight successfully generalizes the model to the new dataset while still retaining knowledge on the old dataset during the continual tuning process. Furthermore, the authors propose a new model-expansion method with instruction grouping strategy based on the phenomenon of instruction reuse, which enables lightweight and limited model expansion on complex and variable datasets. The proposed method could be extended to various Large Foundation Models (LFMs), including MLLMs (LLaVA-7B, LLaVA-13B), LLMs (T5). The authors conduct comprehensive experiments on Multimodal Continual Instruction Tuning (MCIT) benchmark, which consists of Visual Question Answering, Image Classification, OCR, Knowledge Grounding, Reading Comprehension, Visual Reasoning, Visual Grounding etc. The experimental results demonstrate that this method outperforms the previous baselines in terms of Forgetting and Avg.ACC evaluation metrics and achieves the new SOTA results.

## update after rebuttal
Thanks to the authors for addressing my concerns and providing additional results. I will keep my score.

**Claims And Evidence:**

The paper provides clear and convincing evidence to support the claims made in submission.

**Essential References Not Discussed:**

The authors are suggested to include more continual instruction tuning references for LLM in the Related Work section, such as TRACE: A comprehensive Benchmark for Continual Learning in Large Language Models

**Experimental Designs Or Analyses:**

In 5. Experiments section, the authors demonstrated the advantages of the proposed method by comparing to different baselines on various MLLMs, such as LLaVA-7B, LLaVA-13B, and Qwen VL, with common metrics Forgetting and Avg.ACC. The conducted experiments are soundness, convincingly showing the method's effectiveness.

Although the experimental results are impressive and thorough, further modifications are suggested to improve the content of the paper. Especially with regard to the time and memory consumption of model training, it is recommended to further compare the time consumption caused by adopting the dynamic EMA update with the fixed EMA method, and the memory saving with the method that does not use instruction grouping strategy. Further discussing the above comparisons would enhance the understanding of the method.

The ablation studies in Table 6 are insufficient as they only compares with the fixed EMA weight of 0.99. Introducing more fixed EMA weight would reduce randomness and increase the credibility.

**Methods And Evaluation Criteria:**

The proposed methods and evaluation criteria make sense for the problem or application at hand.

**Other Comments Or Suggestions:**

Some terminologies have not been explained well yet. Provide some explanations and significance of plasticity-stability balance in the introduction, which could increase the readability of the paper.

Some phrasing is not very appropriate, such as "knowledge confusion" in line 31 can be changed to "knowledge interference".

**Other Strengths And Weaknesses:**

# Strengths

Novelty: The paper proposes a novel framework to address the continual instruction tuning problem of Large Foundation Models, and its effectiveness has been validated through extensive experiments, demonstrating strong anti-forgetting ability and continual instruction tuning effects. By combining the balance conditions of stability and plasticity with traditional EMA method, the authors derive the optimal solution from an optimization method perspective. This paper offers new ideas and insights for the continual tuning of large models, which is highly inspiring

The paper conducts experiments across various large foundation models and continual instruction tuning benchmarks. These experiments show that the proposed method demonstrates excellent generalizability ability and can be transferred to much more continual instruction tuning scenarios.

The approach presented in this paper is independent of both model architecture and dataset. Therefore, this method can be easily applied to a wider range of models in practical scenarios, and further solves the catastrophic forgetting problem that exists among them.

This paper provides the optimal EMA weight to achieve the plasticity-stability balance from an optimization perspective using the Lagrange multiplier method. Furthermore, by utilizing two optimization methods, the computational load and complexity of the approach are further reduced. The mathematical proofs in the paper are rigorous and reliable, providing a strong theoretical guarantee for the method.

# Weaknesses
Experiments: Lack of more comparative experiments (See Experimental Designs Or Analyses
) and more supplementary experiments (See Supplementary Material).

References: Literature introduction on LLM continual instruction tuning is insufficient (See Essential References Not Discussed).

**Questions For Authors:**

The authors demonstrated in the paper that the dynamic EMA weight may exceed the range of 0-1 (in line 244-246). Therefore, providing the values of EMA weights at each iteration on a specific dataset (such as TextQA) would provide more profound insights.

**Relation To Broader Scientific Literature:**

Compared with other continual instruction tuning methods, this paper makes key contributions in the context of LLMs

**Theoretical Claims:**

The proofs of the deduction process of the optimal EMA weight and two approximate processes are thoroughly checked. Additionally, the complete proof process and further optimization process in the supplementary materials are also been carefully examined. All processes are accurate and convincing.

---

> ### Author Rebuttal · Authors · 2025-03-31
>
> Dear reviewer pe71, thanks for your valuable suggestions. Here are our responses:
>
> **Response 1 (Resource Consumption)**: We compare the time consumption between the dynamic EMA update and the fixed EMA method under same experimental settings (adopting LLaVA-7B). We measure the time consumption for each task on 4-NVIDIA H100 GPUs. The results are shown as:
>
> |Method|ScienceQA|TextVQA|ImageNet|GQA|VizWiz|Grounding|VQAv2|OCRVQA|
> |:---------:|:-------:|:-----:|:------:|:-----:|:----:|:-------:|:----:|:-----:|
> |Dynamic EMA|6 min|25 min|69 min|108 min|12 min|80 min|80 min|112 min|
> |Fixed EMA|6 min|23 min|62 min|100 min|11 min|76 min|75 min|102 min|
>
> We observe that our dynamic EMA update method requires a total of 492 minutes for training across eight tasks. In comparison, the fixed EMA method consumes 455 minutes, indicating that **our method only incurs an 8% increase in training time (37 minutes)**. However, as demonstrated in our experiments, the dynamic EMA update method significantly outperforms the fixed EMA method.
>
> Additionally, we compare the memory consumption of our method with that of approaches that do not use the instruction grouping strategy. Specifically, our method introduces two additional components: one for instruction groups and another for the corresponding LoRA parameters of different instruction groups. Compared to the LoRA parameters, which occupy approximately 1 GB of memory, the saving load for instruction groups—comprising small sentences of natural language text—is negligible, due to requiring only around 1 KB. For simplicity, we define a complete set of LoRA parameters inserted into the LLM as a unit of 1. Under this definition, the method that does not use the instruction grouping strategy shares a single set of LoRA across all tasks, resulting in a quantified memory saving factor of ×1. Our method, using Origin Instruction Type as an example, extends the LoRA parameters across four groups, leading to a quantified memory saving factor of ×4. Given that LoRA is a type of Parameter-Efficient Fine-Tunings (PEFTs), even with a fourfold increase in storage, the total memory load remains smaller compared to the substantial storage demands of LLMs, which often exceed tens of GB [1].
>
> In summary, while adopting the dynamic EMA update and instruction grouping strategy introduces additional time and memory consumption, these costs remain manageable and are significantly outweighed by the substantial performance improvements they promote.
>
> [1] Vicuna: An open-source chatbot impressing gpt-4 with 90%* chatgpt quality.
>
> **Response 2 (Fixed EMA)**: Please kindly refer to the **Response 1** for the same question of Reviewer 9qib.
>
> **Response 3 (Instruction Grouping Results on LLM)**: Please kindly refer to the **Response 6** for the same question of Reviewer 4GQq.
>
> **Response 4 (Reference)**: We have incorporated this reference into the Related Work section and discussed it as “After that, TRACE, another continual instruction tuning benchmark is designed to evaluate the general ability, instruction following and safety for LLMs”.
>
> **Response 5 (Terminology)**: We have added clearer explanation and significance of the “plasticity-stability balance” in the Introduction section as “the trade-off dilemma for models to balance the new task learning and old task storing, which is the core of continual instruction tuning”. Besides that, we have checked the whole manuscript to revise other terminologies that have not been explained well.
>
> **Response 6 (Expression)**: We have changed "knowledge confusion" in line 31 to "knowledge interference", which is a more precise and widely accepted term. Besides that, we have checked the whole manuscript to revise other unsuitable phrases.
>
> **Response 7 (Dynamic EMA)**: We have summarized the values of EMA weights at each iteration on TextQA dataset with LLaVA-7B backbone and Origin instruction type. Due to the rebuttal space limitation, we provide the dynamic EMA weight in the first 10 epochs for the trainable parameters in the first three layers.
>
> |Layers|Epoch 1|Epoch 2|Epoch 3|Epoch 4|Epoch 5|Epoch 6|Epoch 7|Epoch 8 |Epoch 9|Epoch 10|
> |:----:|:-----:|:-----:|:------:|:-----:|:-----:|:------:|:-----:|:------:|:-----:|:------:|
> |1|0.03|0.65|0.85|0.91|0.93|**-0.28**|0.97|0.95|0.97|0.94|
> |2|0.41|0.74|**-0.07**|0.93|0.95|0.97|0.98|0.98|0.94|0.95|
> |3|0.75|0.88|0.84|0.96|0.97|0.98|0.96|**-0.33**|0.96|0.92|
>
> As we can see that the dynamic EMA weight can really exceed the range of 0-1, which demonstrates our theoretical deduction.
>
> Thank you for taking the time to review our manuscript and provide such detailed and constructive suggestion. Your insights have significantly improved our paper, making it much stronger overall. We have carefully considered all your suggestions and made the necessary revisions to ensure the research is more precise and accessible. We deeply appreciate your contribution to enhancing the quality of our work.

---

### Official Review · Reviewer_cghr · 2025-03-14

**Overall Recommendation:** 3

**Summary:**

This paper presents a novel approach to Continual Instruction Tuning for vision-language models, addressing the problem of catastrophic forgetting. The proposed method is built on Exponential Moving Average (EMA)-based updates, dynamically adjusting weight balance based on Taylor expansion and Lagrange multiplier optimization. Additionally, the paper introduces an instruction grouping strategy to minimize redundancy and improve parameter efficiency. The experimental results demonstrate that the method significantly reduces forgetting and enhances continual instruction tuning performance across various vision-language instruction tuning benchmarks.

## update after rebuttal
My original assessment was already supportive, so I will maintain it.

**Claims And Evidence:**

Yes

**Essential References Not Discussed:**

No

**Experimental Designs Or Analyses:**

Yes

**Methods And Evaluation Criteria:**

Yes

**Other Comments Or Suggestions:**

I don’t have additional comments.

**Other Strengths And Weaknesses:**

Strengths:
1. The paper presents a strong theoretical basis for its EMA-weighted update strategy, providing clear mathematical derivations.

Weaknesses:
1. The proposed method provides a general improvement to EMA-based approaches and is not limited to vision-language tasks. However, experiments are conducted solely on vision-language tasks. Additional experiments across diverse task settings would better validate the method's effectiveness.
2. No impact statement is found.

**Questions For Authors:**

1. Can the authors provide more details about the multi-task setup listed in Table 1 and explain why its performance is unexpectedly low, given that the multi-task setting typically serves as a theoretical upper bound for continual learning?

**Relation To Broader Scientific Literature:**

This paper improves widely used EMA-based methods for mitigating forgetting via a dynamic weighting strategy based on theoretical analysis.

**Theoretical Claims:**

Yes

---

> ### Author Rebuttal · Authors · 2025-03-31
>
> Dear reviewer cghr, thanks for your valuable suggestions. Here are our responses:
>
> **Response 1 (Additional Experiments)**: To address your concerns, we set two diverse task settings: Conventional **Image Continual Classification** and **NLP Continual Classification** to further validate the effectiveness of our method.
>
> (1). We choose two **Prompt Continual Learning (PCL)** methods based on ViT backbone, namely L2P [1] and DualPrompt [2]. The **Image Continual Classification Benchmark** is dividing a whole image classification dataset into several splits and each split is seen as a task. We freeze the ViT backbone and only train the prompts. Based on the original L2P and DualPrompt baselines, we integrate our dynamical EMA update method. The continual learning dataset is 10-Split-CIFAR100 with **Class Incremental Learning (CIL)** setting. The experimental results are shown as:
>
> |**Method**|**Avg.ACC**|**Forgetting**|
> |:-----------------:|:---------:|:------------:|
> |L2P|83.77|6.63|
> |**L2P-Ours**|**86.47**|**4.83**|
> |DualPrompt|86.50|5.77|
> |**DualPrompt-Ours**|**88.47**|**3.19**|
>
> In conclusion, our method exhibits superior Avg. ACC and enhanced anti-forgetting performance compared to the original L2P and DualPrompt.
>
> (2). We implement our method on the **NLP Continual Classification Benchmark** (MRCC->SST-2->Sentiment140) with **Task Incremental Learning (TIL)** setting, based on the LLM (Mistral 7B [3]). The **NLP Continual Classification** is selecting multiple NLP classification datasets and each dataset is seen as a task. We freeze the LLM and insert trainable LoRA paradigm into each layer of the LLM respectively. For baseline comparisons, we adopt the original LoRA and one state-of-the-art continual learning method, CurLoRA [4]. The experimental results are shown as:
>
> |**Method**|**Avg.ACC**|**Forgetting**|
> |:--------: |:---------:|:------------:|
> |LoRA|60.00|19.00|
> |CurLoRA|82.00|0.00|
> |**Ours**|**88.00**|**0.00**|
>
> It can be seen that our method exhibits comparable stability while achieving superior plasticity in LLM continual tuning tasks compared to the state-of-the-art method in [4]. Notably, both CurLoRA and our method attain zero forgetting due to the simple TIL setting (knowing the task identifier in the testing stage).
>
> The above experiments demonstrate that **our method can perform well across diverse task settings** and greatly improve the baselines’ performance. These findings further indicate that our method has strong generalization capabilities.
>
> [1] Learning to prompt for continual learning.
>
> [2] Dualprompt: Complementary prompting for rehearsal-free continual learning.
>
> [3] Mistral 7b.
>
> [4] Curlora: Stable llm continual fine-tuning and catastrophic forgetting mitigation.
>
> **Response 2 (Impact Statement)**: Thanks for your kindly reminding. We apologize for the absence of the impact statement. In the revised version, we have added an Impact Statement section to discuss how our approach contributes to continual instruction tuning in LFMs. Specifically, we emphasize that:
>
> 1. This paper presents work whose goal is to advance the field of continual instruction tuning and mitigate catastrophic forgetting in LFMs.
>
> 2. Its relevance to real-world applications, such as lifelong AI assistants and MLLM continual evolution.
>
> We sincerely hope this addition can provide a clearer perspective on the significance and practical implications of our work.
>
> **Response 3 (Multi-Task)**: (1). The multi-task results presented in Table 1 of our paper are directly from the original CoIN paper (Table 2 in [5]). To maintain the authority of the experimental results, we choose to cite the multi-task results from the original report as a reference. (2). The authors of CoIN found that the performance of multi-task is not always higher than that of LoRA continual learning (shown in the following Table, copied from [5]). They explain that “unlike traditional continual learning, where the multi-task model often serves as the upper bound, in CoIN, the performance of the multi-task model is not the best due to the influence of task gaps”. Notably, this phenomenon is consistent with our results. (3). Our idea is that the theoretical upper bound could be understood as the ideal conditions with zero-forgetting, which is represented by the New.ACC metric (The result of training on the single task). While how to choose the upper-bound in continual instruction tuning is still a valuable research topic.
>
> |**Method**|**Backbone**|**Avg.ACC**|
> |:--------:|:--------:|:---------:|
> |LoRA|QWen|41.23|
> |Multi-Task|QWen|41.87|
> |LoRA|MiniGPT|25.45|
> |Multi-Task|MiniGPT|21.50|
>
> [5] Coin: A benchmark of continual instruction tuning for multimodel large language model.
>
> We sincerely appreciate your careful review of our paper. Your valuable suggestions have greatly enhanced the quality of our manuscript by pointing out areas that required improvement. We are truly grateful for the time and effort you dedicated to improving our work.

---

> > ### Comment · Reviewer_cghr · 2025-04-09
> >
> > Thanks to the authors for addressing my concerns and providing additional results. I will keep my score.

---

> > > ### Author Response · Authors · 2025-04-09
> > >
> > > Thanks for your careful review of our paper. We appreciate the effort and time you have spent on improving our work.

---

### Decision · Program_Chairs · 2025-05-01

**Decision:**

Accept (poster)

**Comment:**

The paper received unanimously positive reviews from the reviewers. The proposed solution is simple yet effective and is validated with several strong experimental results. Most of weaknesses pointed out by the reviewers were on suggesting more experimental results as well as some writing issues. AC recommends the authors to include the additional experimental results in the final version to solidify the paper. The recommendation is Accept.